# Quantifying hierarchy and dynamics in US faculty hiring and retention

K. Hunter Wapman[1✉], Sam Zhang[2], Aaron Clauset[1,3,4] & Daniel B. Larremore[1,3✉]

Faculty hiring and retention determine the composition of the US academic workforce and directly shape educational outcomes[1], careers[2], the development and spread of ideas[3] and research priorities[4,5]. However, hiring and retention are dynamic, reflecting societal and academic priorities, generational turnover and efforts to diversify the professoriate along gender[6–8], racial[9] and socioeconomic[10] lines. A comprehensive study of the structure and dynamics of the US professoriate would elucidate the effects of these efforts and the processes that shape scholarship more broadly. Here we analyse the academic employment and doctoral education of tenure-track faculty at all PhD-granting US universities over the decade 2011–2020, quantifying stark inequalities in faculty production, prestige, retention and gender. Our analyses show universal inequalities in which a small minority of universities supply a large majority of faculty across fields, exacerbated by patterns of attrition and reflecting steep hierarchies of prestige. We identify markedly higher attrition rates among faculty trained outside the United States or employed by their doctoral university. Our results indicate that gains in women's representation over this decade result from demographic turnover and earlier changes made to hiring, and are unlikely to lead to long-term gender parity in most fields. These analyses quantify the dynamics of US faculty hiring and retention, and will support efforts to improve the organization, composition and scholarship of the US academic workforce.

Prestige plays a central role in structuring the US professoriate. Analyses of faculty hiring networks, which map who hires whose graduates as faculty, show unambiguously in multiple fields that prestigious departments supply an outsized proportion of faculty, regardless of whether prestige is measured by an extrinsic ranking or reputation scheme[11–13] or derived from the structure of the faculty hiring network itself[14–29]. Prestigious departments also exhibit 'social closure'[15] by excluding those who lack prestige, facilitated by relatively stable hierarchies over time, both empirically[17] and in mathematical models of self-reinforcing network dynamics[30,31].

These observations are important because of the broad impacts of prestige itself. Prestigious affiliations improve paper acceptance rates in single- versus double-anonymous review[32]; faculty at prestigious universities have more resources and write more papers[33,34], receive more citations and attention[35–37] and win more awards[38,39]; and graduates of more prestigious universities experience greater growth in wages in the years immediately after graduating[40]. Furthermore, the vast majority of faculty are employed by departments less prestigious than those at which they were trained[27], making prestigious departments central in the spread of ideas[3] and academic norms and culture more broadly.

Less well studied are the processes of attrition that, together with hiring, shape the data underpinning the analyses reviewed above. Evidence suggests that women in science and engineering (but not mathematics) and foreign-born faculty leave the academy in mid-career at higher rates than do men[41] and US-born[42] faculty, respectively, making

clear the fact that the US professoriate is structured by more than just prestige. These processes are particularly important in light of clear evidence that the topics studied by faculty depend not only on their field of study, but also on their (intersecting) identities[43].

However, the difficulty of assembling comprehensive data on US faculty across fields, across universities and over time has limited analyses and comparisons, leaving it unclear how much of the observed patterns and differences are universal, vary by field or are driven by current or past hiring or attrition. Less visible but just as important are the inherent limitations of focusing only on the placement of faculty within the US system, to the exclusion of US faculty trained abroad. A broad cross-disciplinary understanding of academic hierarchies and their relationship to persistent social and epistemic inequalities would inform empirically anchored policies aimed at accelerating scientific discovery or diversifying the professoriate.

## Data and approach

Our analysis examines tenured or tenure-track faculty employed in the years 2011–2020 at 368 PhD-granting universities in the United States, each of whom is annotated by their doctoral university, year of doctorate, faculty rank and gender. To be included in our analysis, a professor must be a member of the tenured or tenure-track faculty at a department that appears in the majority of sampled years, which yields $n = 295,089$ faculty in 10,612 departments.

[1]Department of Computer Science, University of Colorado Boulder, Boulder, CO, USA. [2]Department of Applied Mathematics, University of Colorado Boulder, Boulder, CO, USA. [3]BioFrontiers Institute, University of Colorado Boulder, Boulder, CO, USA. [4]Santa Fe Institute, Santa Fe, NM, USA. ✉e-mail: hunter.wapman@colorado.edu; daniel.larremore@colorado.edu

This dataset resulted from cleaning and preprocessing a larger US faculty census obtained under a data use agreement with the Academic Analytics Research Center (AARC). To facilitate comparisons of faculty across areas of study, we organized departments into 107 fields (for example, Physics, Ecology) and eight domains (for example, Natural Sciences) (Extended Data Table 1). Field labels, provided in the AARC data, and subsequently hand-checked, are not mutually exclusive, such that 23% of faculty were assigned to multiple fields (for example, members of a Department of Physics and Astronomy were assigned to both Physics and Astronomy). For faculty associated with multiple departments, we restricted our analyses to their primary appointments only. All doctoral universities were manually annotated by country. Self-reported faculty genders were used when available, and otherwise algorithmically annotated (man or woman) on the basis of historical name–gender associations, recognizing that there are expansive identities beyond this limiting binary. These procedures resulted in gender annotations for 85% of records; faculty without name–gender annotations were not included in analyses of gender but were included in all other analyses. Comparing data collected in adjacent years, we also annotated all instances of new hiring, retention and attrition. Data preparation and annotation details can be found in Methods.

To analyse patterns of faculty hiring and exchange among US universities, we created a faculty hiring network for each of the 107 fields, eight domains and for academia as a whole (Methods). In such a network, each node $u$ represents a university, and a directed edge $u \to v$ represents an individual with a doctorate from $u$ who becomes a professor at $v$. Faculty employed at their doctoral universities, so-called self-hires, are represented as self-loops $u \to u$. When aggregating field-level hiring into networks for the eight domains or for academia in toto, we take the union of the constituent fields' edges, which avoids double-counting of faculty rostered in multiple fields. Anonymized data supporting our analyses are freely available (Data availability).

## Pre-eminence of US doctorates

In general, although our data show that US academia largely requires doctoral training, the ecosystem of broad domains and specialized fields exhibits diversity in its credential requirements. Fully 92.7% of all faculty hold doctoral degrees yet only 1% lack a doctorate in Social Sciences compared with 19% in the Humanities (Fig. 1a). Even within the Humanities there is wide variation, with only 7% of remaining faculty lacking a doctorate if one separates out the fields of Theatre (67% non-doctorates), Art History (44%), Music (30%) and English (11%) (Extended Data Fig. 1).

This variation in credentials is paralleled by US faculty trained internationally. Overall, our analysis finds that 11% of US faculty have non-US doctorates yet only 2% of Education faculty received their doctorates internationally compared with 19% of Natural Sciences faculty (Fig. 1a). However, internationally trained faculty primarily receive their training from a limited range of geographical areas, with 35.5% trained in the United Kingdom or Canada compared with just 5.4% from all countries in Africa and the Americas, excluding Canada (Fig. 1b).

Our data suggest that differences in country of doctoral training are not without consequence for the dynamics of the professoriate. Using the 10 years of observations in our data, we identified instances of attrition and estimated the annual per-capita risk of attrition for faculty trained in three groups of countries: Canada and the United Kingdom, the United States, and all others. Those with doctorates from Canada and the United Kingdom ($n = 11,156$) left their faculty positions at statistically indistinguishable rates compared with US-trained faculty ($n = 238,676$) in all 107 fields and eight domains, and at slightly lower rates overall (significance level $\alpha = 0.05$, Benjamini–Hochberg-corrected $\chi^2$ test; Fig. 1c). In stark contrast, those with doctorates from all other countries ($n = 20,689$) left the US tenure track at markedly higher rates overall, in all eight domains and in 39 individual fields (36%), and in no field did such faculty leave at significantly lower rates ($\alpha = 0.05$, Benjamini–Hochberg-corrected $\chi^2$ test; Fig. 1d). We note that our data allow us to consider hypotheses related only to country of doctoral training, not to country of citizenship or birth, leaving open questions about foreign-born yet US-trained faculty[42].

## Universal production inequality

For faculty with US doctorates, we find that academia is characterized by universally extreme inequality in faculty production. Overall, 80% of all domestically trained faculty in our data were trained at just 20.4% of universities. Moreover, the five most common doctoral training universities—UC Berkeley, Harvard, University of Michigan, University of Wisconsin-Madison and Stanford—account for just over one in eight domestically trained faculty (13.8%; Fig. 2a and Extended Data Table 3). Even when disaggregated into domains of study, 80% of faculty were trained at only 19–28% of universities (Fig. 2b).

Our analysis shows that universities that employ more faculty generally also place more of their graduates as faculty elsewhere (Pearson $\rho = 0.76$, two-sided $z$-test $P < 10^{-5}$). Nevertheless, at the level of domains and fields, faculty size alone cannot explain faculty production and placement: in academia as a whole, in all eight domains and in 91 of 107 fields (85%), faculty size and production are from significantly different distributions (Kolmogorov–Smirnov (K-S) test, Benjamini–Hochberg-corrected $P < 10^{-5}$ for academia and domains, $P < 0.01$ for fields), reproducing the findings of a previous analysis of faculty hiring networks in Business, Computer Science and History[27]. For the remaining 16 fields (15%), the hiring of one's own graduates plays a key role: when self-hires are excluded, the distributions of hiring and production of only 12 fields (11%) remain statistically indistinguishable. In other words, inequalities in university or department size do not explain inequalities in faculty production.

The Gini coefficient is a standard way to quantify inequality in a distribution, with $G = 0$ representing perfect equality and $G = 1$ maximal inequality. We find that inequality in faculty production across academia as a whole is both marked ($G = 0.75$) and greater than the inequalities in seven of eight domains. Of those domains, inequality is lowest in Education ($G = 0.67$) and Medicine and Health ($G = 0.67$) and highest in the Humanities ($G = 0.77$). Similarly, inequality in faculty production at the domain level is nearly always greater than production inequality among a domain's constituent fields. For instance, whereas $G = 0.73$ for Engineering as a whole, Gini coefficients for the ten fields within Engineering range from 0.58 to 0.68 and, overall, $G_{domain} > G_{field}$ for 104 of 107 fields (97%; Fig. 3a). Generally, field-level faculty production distributions are heavy tailed and the universities comprising those tails are similar across fields within a given domain and, more broadly, across domains. That is, measurements of inequality in domestic faculty production increase as aggregation or scale expands, because of university-level correlation in faculty production across related fields and domains.

Faculty production inequalities are rooted in hiring but are exacerbated by attrition. Computing the domestic production Gini coefficients separately for newly hired faculty and their sitting colleagues across our longitudinal data frame, we find uniformly larger inequalities for existing faculty in every field, every domain and in academia overall (Fig. 3a). However, cross-sectional Gini coefficients, computed separately for each year of observation, are stable over time, a pattern that rules out a simple cohort effect that would over time draw the Gini coefficients downward towards those of the newly hired faculty (Fig. 3b). Combined, these observations suggest that distributions of faculty production change after hiring in a manner that increases observed inequalities. We tested this hypothesis directly by modelling annual attrition risk as a function of faculty

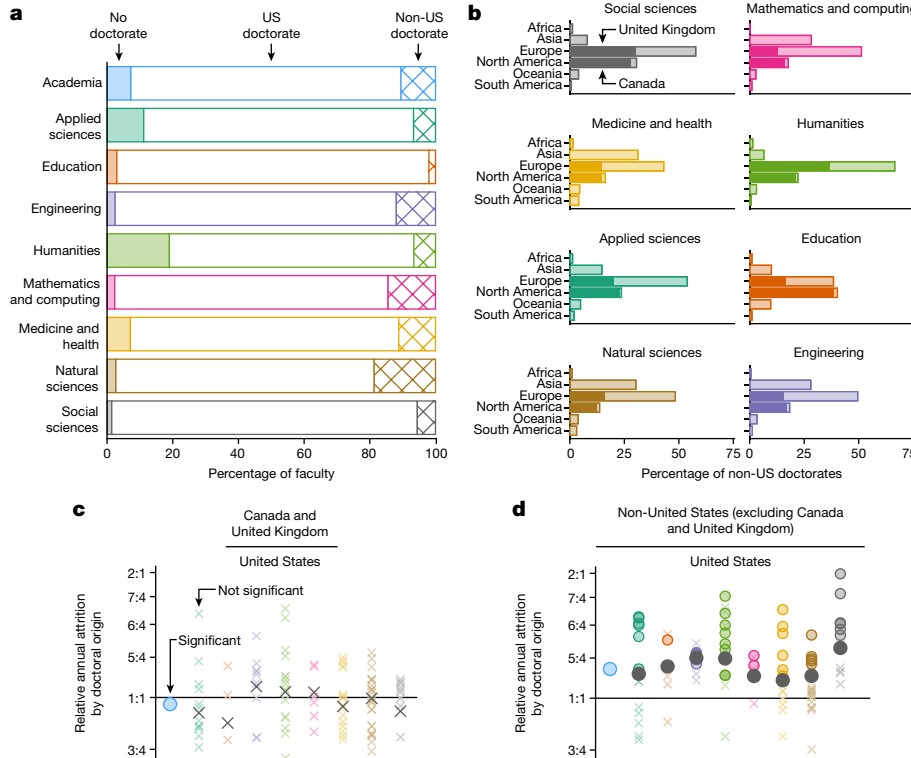

**Fig. 1 | Composition and dynamics of the US professoriate by doctoral training. a**, Degrees of $n = 295,089$ US faculty by domain, and for academia overall, separated by non-doctoral degrees (solid bars), US doctorates (open bars) and non-US doctorates (hatched bars). **b**, Continent of doctorate for $n = 31,845$ faculty with non-US doctorates by domain. Within the Europe and North America bars, darkened regions correspond to faculty from the United Kingdom and Canada, respectively. **c**,**d**, Ratios of average annual attrition risks among faculty with doctorates from Canada and the United Kingdom (**c**) ($n = 11,156$), and from all countries other than Canada, the United Kingdom and the United States (**d**) ($n = 20,689$), versus all US-trained faculty, for each field (colours), domain (grey) and academia (blue), on logarithmic axes. Circles, significantly different from 1.0, $\chi^2$ test, Benjamini–Hochberg-corrected $P < 0.05$; crosses, not significant.

production rank. For academia as a whole, all eight domains and 49 of 107 fields (46%), we find substantially higher rates of attrition among faculty trained at those universities that already produce fewer faculty in the first place (logistic regressions, two-sided $t$-test,

Benjamini–Hochberg-corrected $P < 0.05$). Put differently, most US-trained faculty come from a small number of universities and those who do not are nearly twice as likely to leave the professoriate on an annual basis (Fig. 3c).

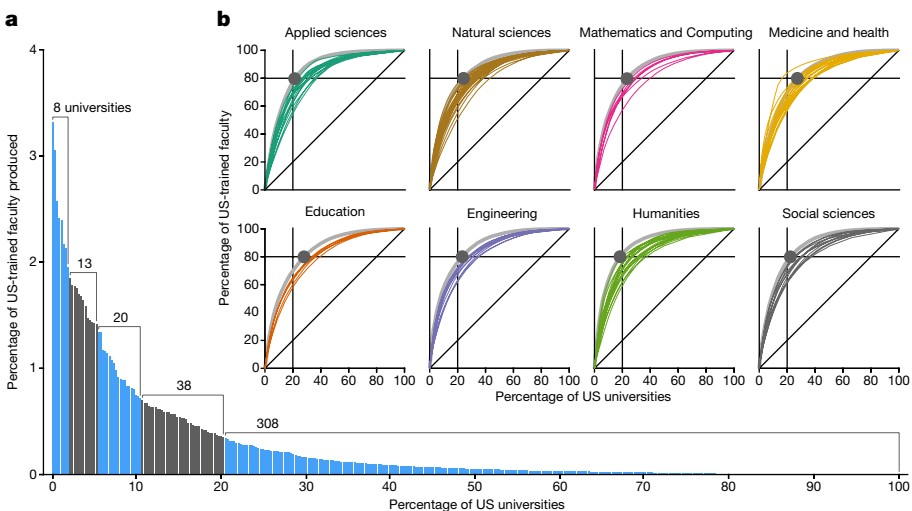

**Fig. 2 | Universal inequality in the production of US-trained faculty. a**, Proportions of US faculty produced by US universities, sorted by production rank, with the university producing the most faculty having a rank of 1 ($n = 238,676$ faculty; $n = 387$ universities). Quintiles of production are highlighted with alternating colours and annotated with the number of universities falling within each. By production, the first quintile contains only eight universities and the bottom contains 308. **b**, Lorenz curves for faculty production at the field level (coloured lines) and at the domain level (grey lines). A point is placed at the site along the domain-level Lorenz curve where 80% of faculty have been produced.

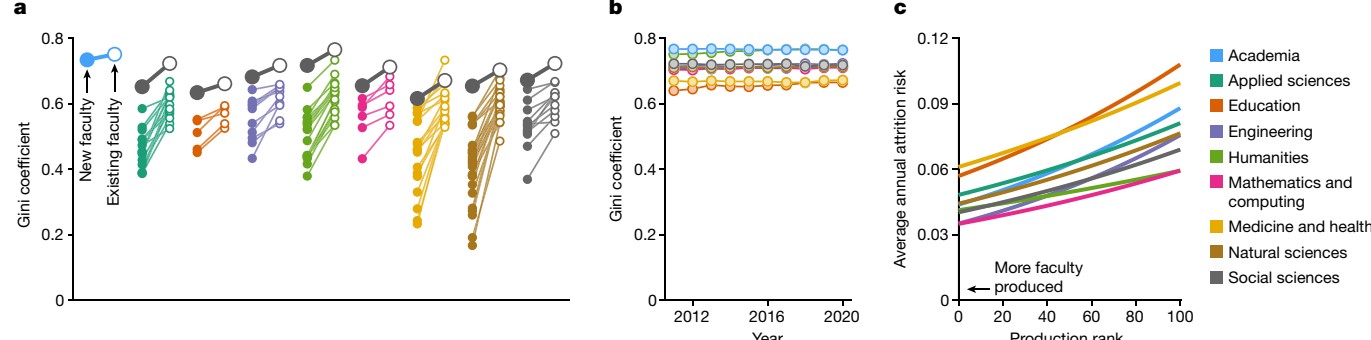

**Fig. 3 | Differential attrition exacerbates inequalities in domestic faculty production. a**, Line segments contrast the faculty production Gini coefficients calculated for newly hired faculty (filled circles; $n$ = 54,100) and for existing faculty (open circles; $n$ = 184,576) for each of the 107 fields (colours), eight domains (grey) and academia as a whole (blue). Line segments are grouped and coloured by domain. **b**, Annual Gini coefficients for academia and for each domain showing strong interyear consistency. **c**, Attrition risk as a

function of university production rank by domain and for academia overall, via logistic regression, showing that university production rank is a significant predictor of annual attrition risk (two-sided $t$-test, Benjamini–Hochberg-corrected $P < 0.05$) such that faculty trained at high-producing universities leave academia at substantially lower rates than those trained at less productive universities. The empirical average annual attrition rates vary around the fitted curves.

## Women on the tenure track

In addition to inequalities in production, our analysis expands on well-documented gender inequalities[8]. Whereas the majority of tenure-track US faculty in our data are men (64%), we find substantial heterogeneity by area of study with moderate change over time. For instance, between 2011 and 2020, women's representation rose from 12.5 to 17.1% among faculty in Engineering and from 55.4 to 58.5% among faculty in Education (Fig. 4a). In fact, women's representation significantly increased in academia overall, in all eight domains and in 80 (75%) of 107 fields (one-sided $z$-test, Benjamini–Hochberg-corrected $P < 0.05$; Fig. 4a). Nursing, a majority-women field, is the single instance in which the representation of women significantly decreased. The representation of women among faculty is thus generally increasing, even as women remain broadly under-represented.

Changes in the overall representation of women over time could be driven by many factors, including demographic changes in new hires between 2011 and 2020 or simply demographic turnover—differences between those entering and those retiring or leaving the professoriate before retirement. Investigating these potential explanations we first found that, between 2011 and 2020, the proportion of women among newly hired faculty did not change significantly in 100 of 107 fields (93%) and significantly decreased in the remaining seven fields (7%).

However, by comparing the inflows of new hires with the outflows of departing faculty over our decade of observation we found that, in academia, all eight domains, and 103 of 107 fields (96%), newly hired faculty were substantially more likely to be women than their departing counterparts (Fig. 4b). This pattern in all-cause attrition is driven by dramatic demographic turnover, with retirement-age faculty skewing heavily towards men (Fig. 4c), implying that the overall increases in women's representation over this period of time (Fig. 4a) are primarily due to changes in faculty hiring that predate our decade of observation. Importantly, the fact that women's representation among new hires has remained flat over the past decade, combined with the observation that newly hired faculty are still more likely to be men (in academia, six of eight domains (75%) and 75 of 107 fields (70%); Fig. 4b), suggests strongly that future gender parity in academia—and especially in Science, Technology, Engineering and Mathematics (STEM) fields—is unlikely without further changes in women's representation among new faculty.

## Self-hiring

Professors who are employed by their doctoral university, called self-hires, account for roughly one in 11 (9.1%) of all US professors in our data (11% of US-trained professors). Whereas these rates remain generally

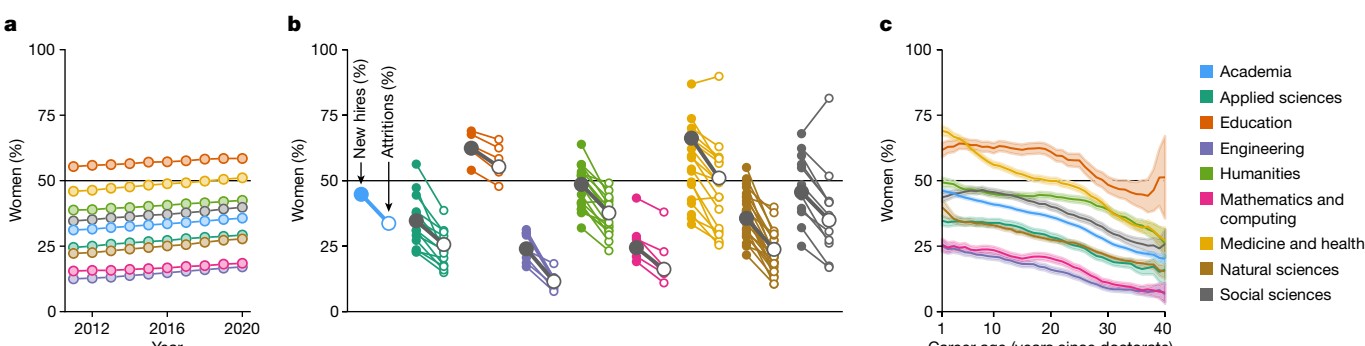

**Fig. 4 | Changes in gender demographics of US faculty. a**, Representation of women over time, coloured by domain and academia ($n$ = 162,408 men, $n$ = 89,429 women). **b**, Line segments contrast percentages of women among newly hired faculty (filled circles; $n$ = 59,007) and women among all-cause attritions (open circles; $n$ = 90,978) for each of the 107 fields (colours), eight

domains (grey) and academia as a whole (blue). Line segments are grouped and coloured by domain. **c**, Representation of women by career age, quantified by years since doctorate, coloured by domain and for academia as a whole. Lines indicate empirical proportions, bands indicate 95% confidence intervals.

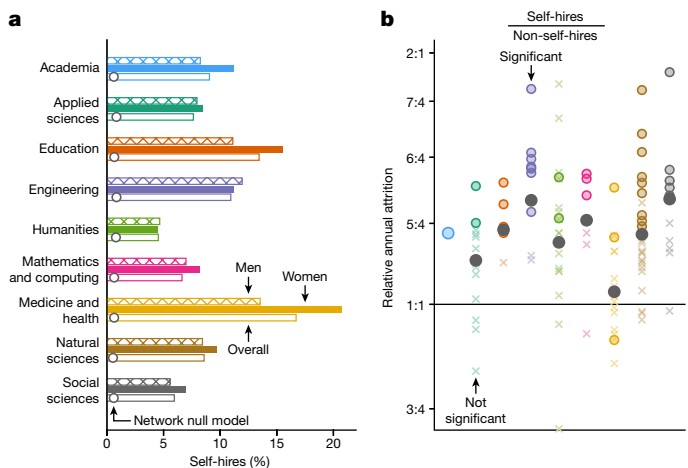

**Fig. 5 | Self-hiring. a**, Self-hiring rates overall (open bars; $n = 295,089$), for women (solid; $n = 89,429$) and for men (hatched; $n = 162,408$), by domain and across academia. Dots overlaid on open bars indicate the expected rate of self-hiring under the network-based null model. **b**, Ratios of average annual attrition risks among self-hires ($n = 26,720$) versus all other faculty ($n = 268,369$) for each field (colours), domain (grey) and academia (blue), on logarithmic axes. Circles, significantly different from 1.0, $\chi^2$ test, Benjamini–Hochberg-corrected $P < 0.05$; crosses, not significant.

low compared with other countries (for example, 36% in Russia[44], 67% in South Africa[29] and 73% in Portugal[45]), they are nevertheless consistently greater than would be expected under a network-based null model that randomizes hiring patterns while keeping faculty production (outflow) and faculty hiring (inflow) fixed[46]. Self-hiring rates were similarly higher than expected across individual fields, ranging from 1.1-fold higher in Theatre to 29.3-fold in Nursing. Self-hiring rates also vary considerably by domain, being lowest in the Humanities (4.5%) and Social Sciences (6.0%) and highest in Medicine and Health (16.7%; Fig. 5a).

Previous work found that women were self-hired at higher rates than men in Computer Science[47]. We find overall that 11.2% of women are self-hires compared with 8.2% of men (two-sided $z$-test for proportions, Benjamini–Hochberg-corrected $P < 10^{-5}$; Fig. 5a). However, this effect is driven by a minority of fields: only 26 (24%) showed differences in self-hiring rates by gender (two-sided $z$-test for proportions, Benjamini–Hochberg-corrected $P < 0.01$), 25 of which featured more frequent self-hiring among women than men. These differences are particularly common in Medicine and Health, where in 12 of 18 fields women are self-hired at significantly higher rates than men.

We also find that self-hires are at greater risk of attrition than non-self-hires. In academia, self-hires in our data leave at 1.2-fold the rate of other faculty and rates are similarly elevated in all eight domains, as well as in 36 of 107 fields (34%; two-sided $z$-test for proportions, Benjamini–Hochberg-corrected $P < 10^{-5}$ for academia, $P < 0.05$ for fields and domains; Fig. 5b). Relative rates of self-hire attrition are highest in Criminal Justice and Criminology and Industrial Engineering, at 1.9- and 1.8-fold the rate of other faculty, respectively. Only in Nursing was the relative rate of self-hire attrition significantly below 1.0 (0.9-fold). It is unclear what drives these differences but, given the ubiquity of self-hired faculty and differential rates of attrition, determining and addressing the causes of this phenomenon would have a wide impact.

## Ubiquitous hierarchies of prestige

If a faculty hiring market were to follow a strict social hierarchy, no university would hire a graduate from a university less prestigious than its own—100% of faculty would hold positions of equal or lower prestige than their doctoral training. The extent to which empirical faculty

hiring networks follow perfect hierarchies has direct implications for academic careers, the mobility of the professoriate and the flow of scientific ideas[3,37]. Treating the flows of faculty between US universities as a network leads to a natural, recursive definition of prestige: a department is prestigious if its graduates are hired by other prestigious departments. We apply the SpringRank algorithm[48] to each faculty hiring network to find, in approximation, an ordering of the nodes (universities) in that network that best aligns with a perfect hierarchy; this ordering represents the inferred hierarchy of prestige.

Faculty hiring networks in the United States exhibit a steep hierarchy in academia and across all domains and fields, with only 5–23% of faculty employed at universities more prestigious than their doctoral university (Fig. 6a,b and Extended Data Table 4). Measured by the extent to which they restrict such upward mobility, these prestige hierarchies are most steep in the Humanities (12% upward mobility) and Mathematics and Computing (13%) and least steep in Medicine and Health (21%; Fig. 6b). We tested whether these steep hierarchies could be a natural consequence of inequalities in faculty production and department size across universities, using a null model in which we randomly rewired the observed hiring networks while preserving out-degree (placements) and in-degree (hires) and ignoring self-loops (self-hires)[46]. For each rewired network we re-ranked nodes using SpringRank and measured induced upward mobility as a test statistic (fraction of up-hierarchy edges; Methods). For academia as a whole, all domains and 94 of 107 fields (88%), empirical networks showed significantly steeper prestige hierarchies than their randomized counterparts (one-sided Benjamini–Hochberg-corrected $P < 0.05$; Fig. 6c and Extended Data Table 5). No field was significantly less steep, although networks in the fields of Pharmacy ($P = 0.88$), Immunology ($P = 0.77$) and Pathology ($P = 0.73$) were less steep than null model randomizations most frequently. In short, the prestige hierarchies that broadly define faculty hiring are universally steep, and often substantially steeper than can be explained by the ubiquitous and large production inequalities.

Inferred prestige ranks of universities are also highly correlated across fields, suggesting that many factors that drive field-level prestige operate at the university level. Among pairwise correlations of university prestige rankings across fields, the overwhelming majority are positive (all but 116 of 12,024) and nearly half (48%) have a correlation >0.7 (Pearson's $\rho$). Fields in Engineering, Mathematics and Computing, and Humanities are particularly mutually correlated whereas the field of Pathology is, on average, the least correlated with others (mean correlation 0.2).

Patterns across field-level 'top-10' most prestigious departments illustrate other aspects of the stark inequalities that define US faculty hiring networks. Among the 1,070 departments that are ranked top-10 in any field, 248 (23.2%) top-10 slots are occupied by departments at just five universities—UC Berkeley, Harvard, Stanford, University of Wisconsin-Madison and Columbia; fully 252 universities (64%) have zero top-10 departments. These findings show that, both within individual fields and across entire domains, faculty placement power is highly concentrated among a small set of universities, complementing the already enormous concentration of faculty production among the same set of universities (Fig. 2). Together, these patterns create network structures characterized by a closely connected core of high-prestige universities that exchange faculty with each other and export faculty to—but rarely import them from—universities in the network periphery (Extended Data Fig. 2).

As a result of both systematic inequality in production and steep social hierarchies, the typical professor is employed at a university that is 18% further down the prestige hierarchy than their doctoral training (Fig. 6a, Extended Data Table 6). Combined with sharply unequal faculty production (Fig. 2), this movement downward in prestige implies that the typical US-trained professor can expect to supervise 2.4-fold fewer future faculty than did their doctoral advisor. At the field level, the typical professor who moves downward descends by between 28%

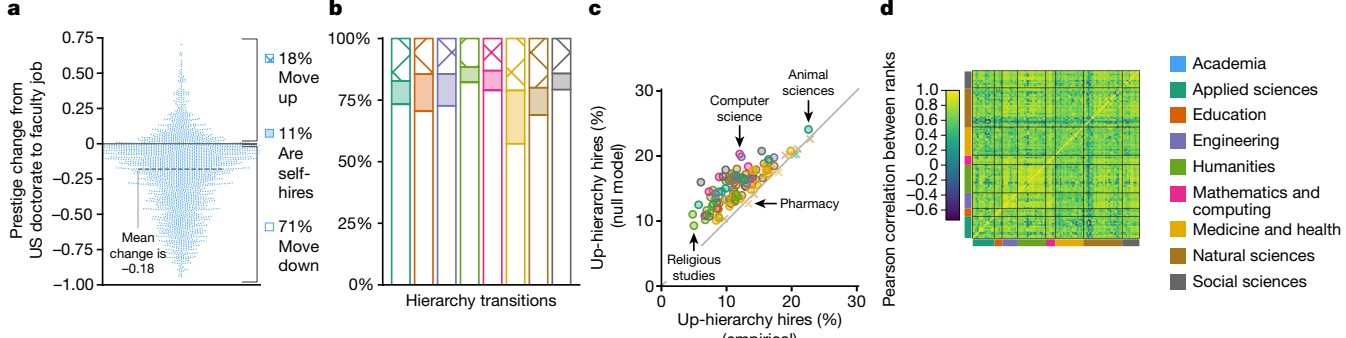

**Fig. 6 | Hierarchies of prestige. a**, Prestige change from doctorate to faculty job in the US faculty hiring network ($n = 238,281$; Methods), with ranks normalized to the unit interval and 1.0 being the most prestigious. The proportions of faculty at universities less prestigious than their doctorate are annotated as 'move down' (open bars), at universities more prestigious than their doctorates as 'move up' (hatched) and at the same university as self-hires (solid). **b**, Rank change among faculty in the US faculty hiring network, by domain, using the same shading scheme as in Fig. 1a. **c**, Comparison between empirical hierarchies and those from 1,000 draws from a null model of randomly rewired hiring networks (Methods), quantified through upward mobility. Fields above the diagonal reference line exhibit steeper hierarchies than can be explained by department size and faculty production inequalities alone. Circles, Benjamini–Hochberg-corrected $P < 0.05$, network null model (Methods); crosses, not significant; no field was significantly less steep than expected. **d**, Heatmap of pairwise Pearson correlations between prestige hierarchies of fields.

(Electrical Engineering) and 46% (Classics) of the prestige hierarchy whereas the typical professor who moves upward, of whom there are very few, ascends by between 6% (Economics) and 26% (Agronomy) of the hierarchy. There was no significant difference in mobility between men and women in 82 of 107 fields, but of the 25 fields in which mobility did differ by gender (two-sided $z$-test for proportions, Benjamini–Hochberg-corrected $P < 0.05$), women were less likely to move down the prestige hierarchy and more likely to be self-hires (Extended Data Table 6); 11 of those 25 fields were within the domain of Medicine and Health. However, we found no significant differences in the magnitudes of upward or downward movements between men and women for all fields (K-S test, Benjamini–Hochberg-corrected $\alpha = 0.05$).

Prestige helps explain more than just the flows of faculty between US universities. For instance, across all domains, our analysis shows that sitting faculty are markedly more likely to be self-hires as prestige increases, yet this relationship is progressively weaker among younger faculty cohorts (Extended Data Fig. 3) and is either attenuated or not significant for new hires (two-sided $t$-test, Benjamini–Hochberg-corrected $\alpha = 0.05$; Extended Data Fig. 4a). By contrast, new hires in all domains are substantially more likely to be trained outside the United States as prestige increases, yet this relationship is either attenuated, not significant or even reversed for sitting faculty (two-sided $t$-test, Benjamini–Hochberg-corrected $\alpha = 0.05$; Extended Data Fig. 4b). Although we observe no common relationship across domains between prestige and gender, both new and existing faculty are more likely to be men as prestige increases for academia as a whole (two-sided $t$-test, Benjamini–Hochberg-corrected $P < 0.05$; Extended Data Fig. 4c). Together, these observations suggest complicated interactions between prestige and the processes of hiring or retaining women, one's own graduates and graduates from abroad, patterns that complement previously observed effects of prestige on peer review outcomes[49,50] and productivity[34].

## Discussion

As a whole, by domain and by field, US tenure-track faculty hiring is dominated by a small minority of US universities that train a large majority of all faculty and sit atop steep hierarchies of prestige. Just five US universities train more US faculty than all non-US universities combined. As we expand our view from fields to entire domains, inequalities in faculty production further increase, reflecting elite universities' positions at or near the top of multiple correlated prestige hierarchies across fields. In principle, universities are on equal footing as both producers and consumers in the faculty hiring market. However, the observed patterns of faculty hiring indicate that the system is better described as having a universal core–periphery structure, with modest faculty exchange among core universities, substantial faculty export from core to periphery and little importation in the reverse direction or from outside the United States.

Although significant efforts have been made over many years to make faculty hiring practices more inclusive, our analysis suggests that many inequalities at the faculty hiring stage are later magnified by differential rates of attrition. For instance, our analysis showed higher rates of attrition among US faculty who were (1) trained outside the United States, Canada or the United Kingdom, (2) trained at universities that have produced relatively fewer faculty overall and (3) employed at their doctoral alma mater. Combined with our observations of unchanging proportions of these groups over time, these differential attrition rates suggest a dynamic equilibrium of countervailing patterns of hiring and attrition. Identifying the causes of these elevated attrition rates is likely to provide insights and opportunities to improve retention strategies for faculty of all kinds.

Our analyses of the hiring and retention of women faculty point to stalled progress towards equal representation. Whereas women's overall representation has increased steadily across all eight broad domains of study, women nevertheless remain under-represented among new hires in many fields, particularly in STEM, and women's representation among newly hired faculty over the past decade has generally been flat. As a result, the continued increase in women's overall representation can instead be attributed to the disproportionate number of men among retiring faculty, across all domains. Continued increases in women's representation among faculty are therefore unlikely if the past decade's pattern remains stable.

Around one in 11 US professors are employed by their doctoral university. Such high rates of self-hiring across fields and universities are surprising, because academic norms treat self-hiring negatively—for example, it is sometimes called 'academic inbreeding'[51]. Elevated self-hiring rates may indicate an unhealthy academic system[52] because self-hiring restricts the spread of ideas and expertise[3], and many decades of study suggest that it can correlate with lower quantity and quality of scholarship[53,54]. In this light, the sharply elevated rates of self-hiring at elite universities present a puzzle[51], with uncertain epistemological consequences, yet these trends seem to be driven less by recent new hires and more by attrition or hiring patterns preceding

our decade of observation. Overall, high rates of self-hiring persist in spite of (not because of) differential rates of attrition, with self-hires leaving US academia at higher rates in most fields, all domains and academia overall.

Our analyses describe system-wide patterns and trends, and hence say little about individual faculty experiences or the causal factors that predict the outcomes of individual faculty placements in the US academic system[55]. At best, our results provide statistical estimates for the direction and distance of faculty placements up or down a field's prestige hierarchy, and they should not be used to inform or shape expectations of real hiring decisions. In other words, even though there are clear and strong patterns at the system level, the considerable variance in outcomes at the individual level shows that pedigree is not destiny.

One limitation of the present work is that, although doctoral universities were known, doctoral departments were not. Hence, our estimates of self-hiring rates reflect faculty employed by any department at their doctoral university, but not necessarily by their doctoral department. Our analyses therefore estimate only upper bounds on department-level self-hiring. Similarly, our estimates of production and prestige inequalities in individual fields reflect the volume and power of universities placing faculty into those fields, but not necessarily the volume of graduates produced by those fields or the related fields into which they may be hired[26].

Our data also lack self-identified demographic characteristics and national origin, which limits the conclusions we may draw about the interaction between faculty hiring and representation by race, gender, socioeconomic background and nationality, and any intersectional analyses thereof. For instance, whereas we observe that faculty trained outside the United States constitute 2–19% of US faculty across domains, the fraction of US faculty born outside the United States is considerably higher[42]. Given our identification of markedly higher attrition rates for faculty trained outside the United States, Canada and the United Kingdom, an investigation of attrition by national origin could help identify its causes and address its differential impacts. Our approach also relies on cultural associations between name and binary man–woman genders, leaving the study of self-identified and more expansive identities, as well as intersectional representation more broadly, as open lines of enquiry.

Although our analysis shows that the clear cross-sectional patterns in faculty demographics and hiring networks are shaped by complex and evolving patterns of hiring and attrition alike, our analysis does not causally identify the mechanisms responsible. Our observations of clustered patterns among fields within the same domain suggest a role for domain-level macrocultures[56]. Strong correlations between a university's ranks across different fields may indicate status signalling[57], the impacts of elite universities' resources on individuals' productivity and prominence[34], or other factors entirely. And, clear cohort effects—particularly in the representation of women—show non-stationarity in the patterns we observe and in the latent factors that drive them. Critically, future progress in understanding the causal factors shaping the US professoriate must investigate the factors that drive differential attrition, including those related to social identity, doctoral training (both abroad and domestically) and university of employment. Understanding the underlying causes of these differential attrition rates would surely inform efforts and policies aimed at mitigating social inequalities by improving equity and representation, which is likely to shape what discoveries are made and who makes them.

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

## Methods

### Data preparation overview

The data used in our analyses are based on a census of the US academic market obtained under a data use agreement with AARC. That unprocessed dataset consisted of the employment records of all tenured or tenure-track faculty at all 392 doctoral-degree-granting universities in the United States for each year between 2011 and 2020, as well as records of those faculty members' most advanced degree. We cleaned, annotated and preprocessed that unprocessed dataset to ensure consistency and robustness of our measurements, resulting in the data used in our analyses.

Cleaning the original dataset involved nine steps, which were performed sequentially. After cleaning, we augmented the processed dataset with two pieces of extra information to enable further analyses of faculty and universities, by annotating the country of each university and the gender of each professor. The nine preparation steps and two annotation steps are described below.

### Data preparation steps

The first step in preparing the dataset was to de-duplicate degree-granting universities. These universities are in our data either because they were 'employing' universities covered by the AARC sample frame (all tenure-track faculty of US PhD-granting universities) or because they were 'producing' universities at which one or more faculty members in the AARC sample frame obtained their terminal degree (university, degree, year). Producing universities include those based outside the United States and those that do not grant PhDs. Thus, due to the AARC sample frame, all employing universities are US-based and PhD granting, and this set of 392 universities did not require preprocessing. On the other hand, producing universities—those where one or more employed faculty earned a degree—may or may not be PhD granting and may or may not be located in the United States.

Producing universities were cleaned by hand: instances in which single universities were represented in multiple ways ('University of Oxford' and 'Keble College', for example) were de-duplicated and, in the rare instances in which a degree referenced an unidentifiable university ('Medical University, England', for example), the degrees associated with that 'university' were removed but the individuals holding those degrees were not removed.

The second step in preparing the dataset was to clean faculty members' degrees. Terminal degrees are recorded for 98.2% of faculty in the unprocessed data: 5.7% of these degrees are not doctorates (5.3% are Master's degrees and 0.4% are Bachelor's degrees). We treated all doctoral degrees as equivalent—for example, we drew no distinction between a PhD and a D.Phil. We note that faculty without doctorates are distributed unevenly throughout academia, with members in the Humanities and Applied Sciences being least likely to have a doctoral degree (Extended Data Fig. 1).

Faculty without doctorates were included in analyses of gender. They were also included in the denominators of self-hiring rate calculations but, possessing no doctorates, they were never considered as potentially self-hires, themselves. Faculty without a doctorate were not included in analyses of production and prestige, which were restricted to faculty with doctorates.

The third step in preparing the dataset was to identify and de-duplicate departments. We ensured that no department was represented multiple different ways, by collapsing records due to (1) multiple representations of the same name (for example, 'Computer Science Department' versus 'Department of Computer Science') and (2) departmental renaming (for example, 'USC School of Engineering' versus 'USC Viterbi School of Engineering'). Although rare instances of the dissolution or creation of departments were observed, we restricted analyses that did not consider time to those departments for which data were available for a majority of years between 2011 and 2020, and restricted longitudinal analyses to only those departments for which data were available for all years.

The fourth step in preparing the dataset was to annotate each department according to a two-level taxonomy based on the field (fine scale) and domain (coarse scale) of its focus. This taxonomy allowed us to analyse faculty hiring at both levels, and to compare patterns between levels. Extended Data Table 1 contains a complete list of fields and domains.

Most departments received just one annotation, but some received multiple annotations due to their interdisciplinarity. This choice was intentional, because the composition of faculty in a 'Department of Physics and Astronomy' is relevant to questions focused on the composition of both ('Physics, Natural Sciences') and ('Astronomy, Natural Sciences'). On the basis of this premise, we include both (or all) appropriate annotations for departments. For instance, the above hypothetical department and its faculty would be included in both Physics and Astronomy analyses. The basic unit of data in our analyses is therefore the individual–discipline pair. A focus on the individual would be preferable, but would require taxonomy annotations of individuals rather than departments—information we do not have. Furthermore, many individuals are likely to consider themselves to be members of multiple disciplines.

Whenever a university had multiple departments within the same field, those departments were considered as one unit. To illustrate how this was done, consider the seven departments of Carnegie Mellon's School of Computer Science. All seven departments were annotated as Computer Science and treated together in analyses of Computer Science.

Some fields have the potential to conceptually belong to multiple domains. For example, Computer Engineering could be reasonably included in the domain of either Formal Sciences (which includes Computer Science) or Engineering (which includes Electrical Engineering). Similarly, Educational Psychology could be reasonably included in the domain of Education or of Social Sciences. In these instances, we associated each such field with the domain that maximized the fraction of faculty whose doctoral university had a department in that domain. In other words, we matched fields with domains using the heuristic that fields are best associated with the domains in which their faculty are most likely to have been trained.

The fifth step in preparing the dataset was to remove inconsistent employment records. Rarely, faculty in the dataset seem to be employed at multiple universities in the same year. These cases represent situations in which a professor made a mid-career move and the university from which they moved failed to remove that professor from their public-facing records. We removed such spurious and residual records for only the conflicting years, and left the records of employment preceding such mid-career moves unaltered. This removed only 0.23% of employment records.

The sixth step in preparing the dataset was to impute missing employment records. Rarely, faculty disappear from the dataset only to later reappear in the department they left. We considered these to be spurious 'departures', and imputed employment records for the missing years using the rank held by the faculty before becoming absent from the data. Employment records were not imputed if they were associated with a department that did not have any employment records in the given year. Imputations affected 1.3% of employment records and 4.7% of faculty.

The seventh step in preparing the dataset was to exclude non-primary appointments such as professors' associations or courtesy/emeritus appointments with multiple departments. We identified primary appointments by making the following two assumptions. First, if a professor was observed to have just one appointment in a particular year, then that was their primary appointment for that year—as well as for any other year in which they held that appointment (including years with multiple observed appointments). This corresponds to a heuristic that faculty should appear on the roster of their primary unit

before appearing on non-primary rosters. Second, if a professor was observed to have appointments in multiple units, and a promotion (for example, from Assistant Professor to Associate Professor) was observed in one unit's roster but not in another's, it was assumed that the non-updating unit is not a primary appointment. This corresponds to a heuristic that, if units vary in when they report promotions, it is more likely that the primary unit is updated first and thus units that update more slowly are non-primary.

Primary appointments could not be identified for 1.2% of faculty, and 5.5% of appointments were classified as non-primary. Field- and domain-level analyses were restricted to primary appointments, but analyses of academia included faculty regardless of whether their primary appointment(s) could be identified, under the assumption that employment in a tenure-track position implies having some primary appointment, identifiable or not.

The eighth step in preparing the dataset was to carefully handle employment records with mid-career moves so that each faculty member was associated with only a single employing university. Mid-career moves do not alter a professor's doctoral university or gender, and so cannot affect measurements such as a discipline's faculty production Gini coefficient, its gender composition or the fraction of faculty within the discipline that holds a degree from outside the United States. However, mid-career moves have the potential to alter a discipline's self-hire rate and the steepness of its prestige hierarchy. This raises important questions for how one should treat mid-career moves when performing calculations that average over our decade of observations—should one analyse the appointment before or the appointment after the move(s)?

First we chose to use, whenever possible, the most recent employing university of each professor. In other words, if a professor was employed at multiple universities between 2011 and 2020, only that university where they were most recently employed was considered. Second, we checked that this choice did not meaningfully affect our analyses of self-hiring or prestige, because 6.9% of faculty made a mid-career move within our sample frame. To evaluate the impact of this choice on self-hiring analyses, we first calculated self-hiring rates on the basis of faculty members' first employing university (that is, their pre-mid-career-move university if they had a mid-career move). We then calculated self-hiring rates on the basis of faculty members' last employing university (that is, their post-mid-career-move university if they had a mid-career move). Comparing these two estimates we found that, across all 107 fields, eight domains and academia, mid-career moves had no significant effect on our measurements of self-hiring rates (two-sided $z$-test for proportions, $\alpha = 0.05$, $n = 295,089$ faculty in both samples). To evaluate the impact of this choice on prestige hierarchies, we first calculated the upward mobility in rank-sorted faculty hiring networks on the basis of faculty members' first employing university (that is, their pre-mid-career move university if they had a mid-career move). We then followed the same procedure but on the basis of faculty members' last employing university (that is, their post-mid-career move university if they had a mid-career move). Comparing these two approaches, we found that mid-career moves did not significantly alter upward mobility in any field or domain (two-sample, two-sided $z$-test for proportions, Benjamini–Hochberg-corrected $\alpha = 0.05$; see Extended Data Table 1 for $n$). At the academia level, taking the most recent university rather than the first university among mid-career moves resulted in 0.7% more upwardly mobile doctorate-to-faculty transitions (two-sample, two-sided $z$-test for proportions, Benjamini–Hochberg-corrected $P < 0.05$, $n = 238,281$ in both samples).

The ninth and final step in preparing the dataset was to exclude departments that were inconsistently sampled. Not all departments in the unprocessed dataset were recorded by the AARC in all years, for reasons outside the control of the research team. To ensure robustness of results, we restricted our analyses that did not consider time to those departments that appeared in a majority of years between 2011 and 2020. This resulted in the removal of 1.8% of employment records, 3.4% of faculty and 9.1% of departments. Additionally, 24 employing universities (6.1%) were excluded by this criterion, most of which were seminaries.

## Annotations

The country of each producing university was determined by hand. First, Amazon Mechanical Turk was used to gather initial annotations. Each university was annotated by two different annotators. Inter-annotator agreement was >99% and disagreements were readily resolved by hand. To ensure no errors, a second pass was completed by the researchers and resulted in no alterations.

Self-identified gender annotations were provided for 6% of faculty in the unprocessed dataset. To annotate the remaining faculty with gender estimates, we used a two-step process based on first and last names. First, complete names were passed to two offline dictionaries: a hand-annotated list of faculty employed at Business, Computer Science and History departments (corresponding to the data used in ref. [27]) and the open-source python package gender-guesser[58]. Both dictionaries responded with one of the following classifications: female, male or unable to classify. Second, for cases in which the dictionaries either disagreed or agreed but were unable to assign a gender to the name, we queried Ethnea[59] and used the gender to which they assigned the name (if any). Using this approach we were able to annotate 85% of faculty with man or woman labels. Faculty whose names could not be associated with a gender were excluded from analyses of gender but included in other analyses. This methodology associates names with binary (man/woman) labels because of technical limitations inherent in name-based gendering methodologies, but we recognize that gender is non-binary. The use of these binary gender labels is not intended to reinforce a gender binary.

## Per-analysis inclusion criteria

The prepared and annotated dataset contained 295,089 individuals employed at 368 universities, and was used as the basis of all of our analyses. In some analyses, further inclusion criteria were applied but with the guiding principle that analyses should be as inclusive as possible and reasonable. For example, analyses of the professoriate by gender considered only faculty with a gender annotation but did not require members to hold a doctorate. Analyses of prestige, on the other hand, considered only those faculty with doctorates from US universities but did not require that faculty have a gender annotation. The aim of these inclusion criteria was to ensure the robustness of results while simultaneously being maximally inclusive. When an analysis fell into more than one of the above categories, inclusion criteria for all categories were applied. For example, when analysing changes in US faculty production over time, inclusion criteria for analyses of both US faculty production and over time were applied.

Some fields and domains were excluded from field- or domain-level analyses, either because they were too small or because they were insufficiently self-contained. Faculty in excluded fields were nevertheless included in domain- and academia-level analyses, and those in excluded domains were nevertheless included in academia-level analyses (Extended Data Table 2).

Two domains were excluded from domain-level analysis: (1) Public Administration and Policy and (2) Journalism, Media and Communications. These domains were excluded because they employed far fewer faculty than other domains, and because their inclusion made domain-level comparisons difficult.

Fields were included in field-level analyses only if (1) at least 25% of universities had a department in that field or (2) the number of faculty with a primary appointment in that field, and who also earned their doctorate from a university that had a department in that field, was ≥500. These requirements were intended to ensure the coherence of

fields for analyses of production and prestige. For information on the number of faculty excluded from field- and domain-level analyses, see Extended Data Table 2.

Analyses of production and prestige included only faculty who hold a US doctorate. Faculty without a doctorate are a small minority of the population in most fields, and were excluded because their degrees are not directly comparable to doctorates. Faculty with non-US doctorates were excluded because the universities that produced them are outside the sample frame.

For all longitudinal analyses, we required departments to be sampled in all years between 2011 and 2020 to ensure consistency in the sample frame. This resulted in the removal of 5.9% of employment records, 7.2% of faculty and 12.6% of departments for those analyses. Additionally, 15 employing universities (4.1%) were excluded by this criterion.

### Identification of new hires

Some analyses required us to divide faculty into two complementary sets: new hires and existing faculty. For analyses that aggregated faculty over our decade of observation, we labelled faculty as new hires if they earned their degree within 4 years of their first recorded employment as faculty. Thus defined there are 59,007 new faculty, making up 20.0% of the faculty in the dataset. Our longitudinal analyses were more strict, such that faculty were labelled as new only in their first observed year of employment, but were considered as existing faculty for each observed year thereafter.

### Identification of attrition and calculation of attrition risk

A professor who leaves academia for any reason constitutes an attrition, including retirement, termination of employment for any reason, acceptance of a position outside our sample frame (for example, in industry, government or a university outside the United States) or death. Our unprocessed data do not allow us to identify reasons for attrition. A professor's last year of employment is considered the year of their attrition when counting attritions over time. Faculty who change disciplines are not considered to be attritions from disciplines they leave. Because attritions in a given year are identified through comparison with employment records in the next, attrition analyses do not include the final year of the sample frame (2020). Faculty were counted as an attrition at most once; a professor who appeared to leave multiple times was considered an attrition only on exiting for the last time.

Attrition risk is defined, for a given set of faculty in a given year, as the probability that each professor in that set failed to appear in the set in the next year—that is, the proportion of observed leaving events among possible leaving events on an annual basis. Thus, all attrition risks as stated in this study are annual per-capita risks of attrition. Average annual attrition risks were formed by counting all attrition events and dividing by the total person-years at risk.

### Faculty hiring networks

Faculty hiring networks represent the directed flows of faculty from their doctoral universities to their employing universities. As such, each node in such a network represents a university and each weighted, directed edge represents the number of professors trained at one university and who are employed at the other. For the purposes of the faculty hiring networks analysed here, we restrict the set of nodes to, at most, those employing universities within the AARC sample frame. This means that nodes representing non-US universities are not included, and therefore the edges that would link them to in-sample universities are also not included. Without loss of generality, we now describe in more precise detail the creation of a particular field's faculty hiring network, but this process applies equivalently for both domains and academia as a whole.

First, universities were included in a field only if they had a unit (for example, a department, or departments) associated with that field. As a result, a university appears in the rankings for a field only if it has a representative unit; without a Department of Botany, a university cannot be ranked in Botany. Second, ranks are identifiable from patterns in faculty hiring only if every unit employs at least one individual in that field who was trained at a unit that also employs faculty in that field. Phrased from the perspective of the faculty hiring network, this requirement amounts to ensuring that the in-degree of every node is at least one. Because the removal of one unit (based on the above requirements) might cause another to fail to meet the requirements, we applied this rule repeatedly until it was satisfied by all units.

The outcome of this network construction process is a weighted, directed multi-graph $A^{(k)}$ such that: (1) the set of nodes $i = 1, 2, \ldots$ represent universities with a department or unit in field $k$. (2) The set of edges represent hiring relationships, such that $A_{ij}^{(k)}$ is an integer count of the number of faculty in field $k$ who graduated from $i$ and are employed at $j$. Thus $A^{(k)}$ is a positive, integer-weighted, non-symmetric, network adjacency matrix for field $k$. (3) The out-degree $d_i^{(k)} = \sum_j A_{ij}^{(k)}$ is greater than or equal to one for every node $i$, meaning that every university has placed at least one graduate in field $k$. (4) The in-degree $d_j^{(k)} = \sum_i A_{ij}^{(k)}$ is greater than or equal to one for every node $j$, meaning that every university has hired at least one graduate from field $k$.

To infer ranks in faculty hiring networks meeting the criteria above, we used the SpringRank algorithm[48] without regularization, producing a scalar embedding of each network's nodes. Node that embeddings were converted to ordinal rank percentiles. (In principle, embeddings may produce ties requiring a rule for tie-breaking when converting to ordinal ranks. However, no ties in SpringRanks were observed in practice).

To determine whether properties of an empirically observed hierarchy in a faculty hiring network could be ascribed to its in-degree sequence (unit sizes) and out-degree sequence (faculty production counts) alone, we generated an ensemble of $n = 1,000$ networks with identical in- and out-degrees that were otherwise entirely random, using a degree-preserving null model called the configuration model[46,60]. We excluded self-hires (that is, self-loops) from randomization in the configuration model for a subtle but methodologically important reason. We observed that self-hires occur at much higher rates in empirical networks than expected under a configuration model. As a result, were we to treat self-hires as links to be randomized, the process of randomization would, itself, increase the number of inter-university hires from which ranks were inferred. Because of the fact that SpringRank (or an alternative algorithm) infers ranks from inter-university hires, but not self-hires, the act of 'randomizing away' self-hires would thus distort ranks, as well as the number of potential edges aligned with (or aligned against) any inferred hierarchy. In short, randomization of self-hires would, in and of itself, distort the null distribution against which we hope to compare, dashing any hope of valid inferences to be drawn from the exercise. We note, with care, that when computing the fraction of hires violating the direction of the hierarchy, either empirically or in the null model, we nevertheless included self-hires in the total number of hires—that is, the denominator of said fraction. These methodological choices follow the considerations of the configuration model 'graph spaces' introduced by Fosdick et al.[46].

### Reporting summary

Further information on research design is available in the Nature Research Reporting Summary linked to this article.

### Data availability

All network data associated with this study and all data contained in Extended Data tables are freely available in machine-readable format at https://doi.org/10.5281/zenodo.6941651. Explorable visualizations

of faculty hiring networks and university ranks are available at https://larremorelab.github.io/us-faculty/. Source data are provided with this paper.

## Code availability

Open-source code related to this study is available at https://doi.org/10.5281/zenodo.6941612.

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

**Acknowledgements** The authors thank A. Morgan, N. LaBerge and C. J. E. Metcalf for valuable feedback, and acknowledge the BioFrontiers Computing Core at the University of Colorado Boulder for providing High Performance Computing resources supported by BioFrontiers IT. This work was supported by an Air Force Office of Scientific Research Award (no. FA9550-19-10329, all authors), by a National Science Foundation Graduate Research Fellowship Award (no. DGE-2040434, S.Z.) and by a National Science Foundation Alan T. Waterman Award (no. SMA-2226343, D.B.L.).

**Author contributions** K.H.W., A.C. and D.B.L. devised the analysis and wrote the manuscript. K.H.W. performed computational modelling and validated the data. K.H.W. and S.Z. processed the data. D.B.L. supervised the project.

**Competing interests** The authors declare no competing interests.

**Additional information**
**Correspondence and requests for materials** should be addressed to K. Hunter Wapman or Daniel B. Larremore.

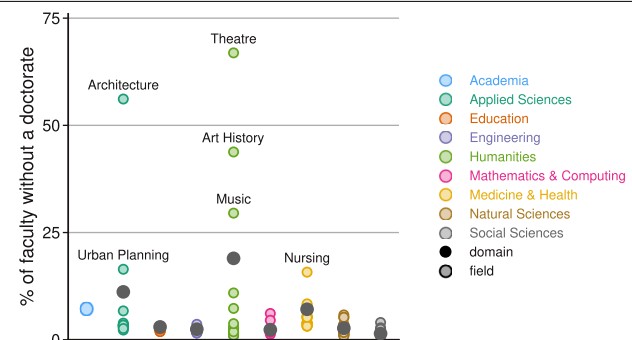

**Extended Data Fig. 1 | Proportions of faculty without doctoral degrees.**
Each transparent circle represents one of 107 fields, coloured and grouped by
domain. Filled grey circles represent domain-level estimates. A single blue
circle (left) represents U.S. academia overall. Fields for which more than 10% of
faculty do not have a doctorate are annotated.

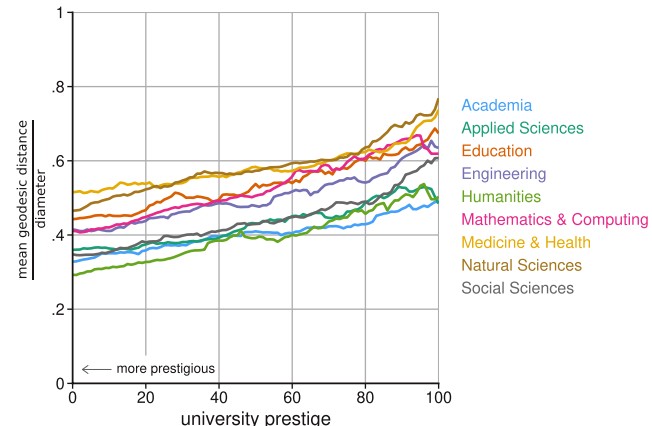

**Extended Data Fig. 2 | University network centrality as a function of prestige.** Lines are coloured by domain, and show the mean geodesic distance through links in the faculty hiring network from the university at that prestige rank to every other university, divided by the diameter of the network. Smaller values toward the left side indicate that more prestigious universities are more centrally located in each faculty hiring network; less prestigious universities are more peripherally positioned. All universities belong to the network's strongly connected component by construction (Methods).

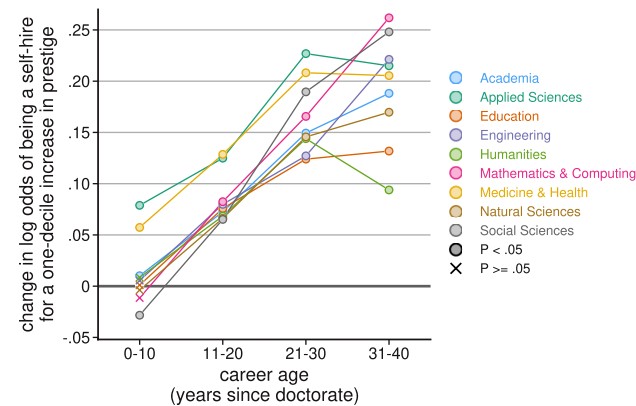

**Extended Data Fig. 3 | Self-hire rates as a function of prestige and career age.** Logistic regression coefficients, expressed as change in log-odds of being a self-hire for a one-decile increase in prestige, stratified by domain (colours) or academia (blue), and by four bins of career age as indicated. Circles, significant by two-sided t-test, Benjamini-Hochberg corrected $p < .05$; crosses, not significant.

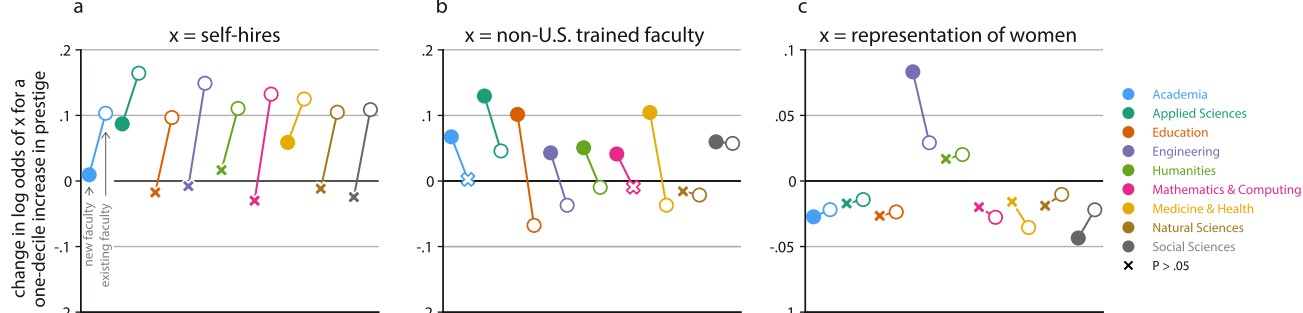

**Extended Data Fig. 4 | Effects of prestige.** Logistic regression coefficients, expressed as a change in log-odds of faculty being a self-hire (a), being a non-U.S. faculty (b), or a woman (c) for a one-decile increase in prestige, stratified by domain (colours) and academia (blue), for newly hired faculty (filled symbols) and for existing faculty (hollow symbols) and connected by a line. Circles, significant (two-sided t-test, Benjamini-Hochberg corrected $p > 0.05$); crosses, not significant. (a) Existing faculty are more likely to be self-hires at more prestigious universities, but this effect attenuates or disappears for new hires, indicating that the positive relationship between self-hiring and prestige is likely driven by attrition. (b) Newly hired faculty are more likely to hold a non-U.S. doctorate than existing faculty. This likely results from higher rates of attrition among faculty with a non-U.S. doctorate (Fig. 1c). (c) We observe no universal relationship across domains between prestige and gender, but both new and existing faculty are somewhat more likely to be men as prestige increases for academia as a whole.

## Extended Data Table 1 | Hierarchical taxonomy of academia

| Domain / Field | n faculty | % of domain | % of academia |
|---|---|---|---|
| **Applied Sciences** | 30,665 | — | 10.4 |
| Accounting | 2995 | 9.8 | |
| Agronomy | 1119 | 3.6 | |
| Animal Sciences | 1913 | 6.2 | |
| Architecture | 2900 | 9.5 | |
| Business Administration | 3175 | 10.4 | |
| Finance | 3275 | 10.7 | |
| Food Science | 1377 | 4.5 | |
| Horticulture | 907 | 3.0 | |
| Management | 5536 | 18.1 | |
| Management Information Sys. | 1882 | 6.1 | |
| Marketing | 2682 | 8.7 | |
| Plant Sciences | 1611 | 5.3 | |
| Soil Science | 1235 | 4.0 | |
| Urban & Regional Planning | 1369 | 4.5 | |
| **Education** | 13,980 | — | 4.7 |
| Counselor Education | 1687 | 12.1 | |
| Curriculum & Instruction | 3548 | 25.4 | |
| Education | 3227 | 23.1 | |
| Education Administration | 3362 | 24.0 | |
| Special Education | 1314 | 9.4 | |
| **Engineering** | 29,443 | — | 10.0 |
| Aerospace Engineering | 2557 | 8.7 | |
| Agricultural Engineering | 2598 | 8.8 | |
| Civil Engineering | 4415 | 15.0 | |
| Electrical Engineering | 7404 | 25.1 | |
| Environmental Engineering | 3619 | 12.3 | |
| Industrial Engineering | 1765 | 6.0 | |
| Materials Engineering | 2517 | 8.5 | |
| Mechanical Engineering | 6317 | 21.5 | |
| Operations Research | 1394 | 4.7 | |
| Systems Engineering | 1357 | 4.6 | |
| **Humanities** | 50,610 | — | 17.2 |
| Art History & Criticism | 3046 | 6.0 | |
| Asian Languages | 641 | 1.3 | |
| Asian Studies | 766 | 1.5 | |
| Classics & Classical Languages | 1603 | 3.2 | |
| Comparative Literature | 753 | 1.5 | |
| English Language & Literature | 8924 | 17.6 | |
| French Language & Literature | 820 | 1.6 | |
| Germanic Languages & Lit. | 725 | 1.4 | |
| History | 7256 | 14.3 | |
| Linguistics | 1191 | 2.4 | |
| Music | 6269 | 12.4 | |
| Near/Mid. Eastern Lang./Cultures | 540 | 1.1 | |
| Philosophy | 3718 | 7.3 | |
| Religious Studies | 1876 | 3.7 | |
| Slavic Languages & Literatures | 534 | 1.1 | |
| Spanish Language & Literature | 1191 | 2.4 | |
| Theatre Literature, History & Crit. | 2106 | 4.2 | |
| Theological Studies | 2131 | 4.2 | |
| **Math & Computing** | 25,969 | — | 8.8 |
| Computer Engineering | 6805 | 26.2 | |
| Computer Science | 8080 | 31.1 | |
| Information Science | 1915 | 7.4 | |
| Information Technology | 1780 | 6.9 | |
| Mathematics | 8921 | 34.4 | |
| Statistics | 3401 | 13.1 | |
| **Medicine & Health** | 54,849 | — | 18.6 |
| Communication Disord.& Sci. | 1282 | 2.3 | |
| Environmental Health Sci. | 1295 | 2.4 | |
| Epidemiology | 2711 | 4.9 | |
| Exercise Sci., Kines., Rehab | 5467 | 10.0 | |
| Genetics | 1266 | 2.3 | |
| Health, Phys. Ed., Recreation | 1442 | 2.6 | |
| Human Dev. & Family Sci. | 2162 | 3.9 | |
| Immunology | 3330 | 6.1 | |
| Nursing | 7931 | 14.5 | |
| Nutrition Sciences | 2161 | 3.9 | |
| Pharmaceutical Sciences | 2568 | 4.7 | |
| Pharmacology | 3260 | 5.9 | |
| Pharmacy | 2069 | 3.8 | |
| Physiology | 3601 | 6.6 | |
| Public Health | 5882 | 10.7 | |
| Social Work | 3653 | 6.7 | |
| Speech & Hearing Sciences | 992 | 1.8 | |
| Veterinary Medical Sciences | 3915 | 7.1 | |
| **Natural Sciences** | 70,791 | — | 24.0 |
| Anatomy | 2109 | 3.0 | |
| Astronomy | 3401 | 4.8 | |
| Atmosph.Sci. & Meteorology | 1549 | 2.2 | |
| Biochemistry | 6419 | 9.1 | |
| Biological Sciences | 8641 | 12.2 | |
| Biomedical Engineering | 2691 | 3.8 | |
| Biophysics | 1565 | 2.2 | |
| Biostatistics | 2069 | 2.9 | |
| Cell Biology | 4260 | 6.0 | |
| Chemical Engineering | 3057 | 4.3 | |
| Chemistry | 7043 | 9.9 | |
| Ecology | 1382 | 2.0 | |
| Entomology | 1121 | 1.6 | |
| Environmental Sciences | 3014 | 4.3 | |
| Evolutionary Biology | 1020 | 1.4 | |
| Forestry & Forest Resources | 1444 | 2.0 | |
| Geology | 4287 | 6.1 | |
| Marine Sciences | 1525 | 2.2 | |
| Microbiology | 4547 | 6.4 | |
| Molecular Biology | 4005 | 5.7 | |
| Natural Resources | 1821 | 2.6 | |
| Neuroscience | 3403 | 4.8 | |
| Pathology | 6530 | 9.2 | |
| Physics | 7678 | 10.8 | |
| Plant Pathology | 1073 | 1.5 | |
| **Social Sciences** | 38,019 | — | 12.9 |
| Agricultural Economics | 1216 | 3.2 | |
| Anthropology | 3862 | 10.2 | |
| Crim. Justice & Criminology | 1585 | 4.2 | |
| Economics | 6052 | 15.9 | |
| Educational Psychology | 1890 | 5.0 | |
| Gender Studies | 648 | 1.7 | |
| Geography | 2228 | 5.9 | |
| International Affairs | 1717 | 4.5 | |
| Political Science | 5710 | 15.0 | |
| Psychology | 8963 | 23.6 | |
| Sociology | 4727 | 12.4 | |

Domains (e.g. Natural Sciences, or Medicine and Health; highlighted, bold italics) contain Fields (e.g. Physics, or Nutritional Sciences). Columns show the number of faculty in each field and domain, with percentages showing the relative proportions of fields in domains, and domains in academia. We note that percentages need not sum to 100, as some faculty appear in multiple fields or domains (Methods). See Data Availability for complete machine-readable data and taxonomy.

**Extended Data Table 2 | Faculty excluded from field- and domain-level analyses**

| Domain | % of domain excluded | % of academia excluded |
|---|---|---|
| Applied Sciences | 10.59% (3358) | |
| Education | 16.01% (2429) | |
| Engineering | 7.7% (2331) | |
| Humanities | 19.77% (10,255) | |
| Journalism, Media, Communication | | 1.93% (5899) |
| Mathematics & Computing | 1.45% (389) | |
| Medicine & Health | 7.87% (4532) | |
| Natural Sciences | 5.47% (4026) | |
| Public Administration & Policy | | 0.56% (1725) |
| Social Sciences | 3.21% (1253) | |

Some fields and domains included in the original AARC data were excluded from field- or domain-level analysis due to their small size. For domains that contained at least one excluded field, a column shows the percentage of faculty employed in that domain who were excluded from field-level analyses. For the two domains that were excluded from domain-level analyses, we show the percentage of faculty employed in academia who were excluded from domain-level analyses (see Methods).

**Extended Data Table 3 | Faculty production ranks by university**

| # | University | % of faculty produced | # | University | % of faculty produced |
|---|---|---|---|---|---|
| 1 | UC Berkeley | 3.32 | 51 | Brown | 0.61 |
| 2 | Harvard | 3.05 | 52 | UMass Amherst | 0.59 |
| 3 | U Michigan | 2.58 | 53 | U Kentucky | 0.59 |
| 4 | U Wisconsin-Madison | 2.41 | 54 | Vanderbilt | 0.58 |
| 5 | Stanford | 2.39 | 55 | UC Santa Barbara | 0.57 |
| 6 | U Illinois Urbana-Champaign | 2.17 | 56 | Carnegie Mellon | 0.57 |
| 7 | MIT | 2.14 | 57 | Georgia Tech | 0.57 |
| 8 | UT Austin | 1.96 | 58 | University at Buffalo (SUNY) | 0.54 |
| 9 | Cornell | 1.85 | 59 | LSU | 0.54 |
| 10 | Columbia | 1.79 | 60 | U Tennessee | 0.54 |
| 11 | Yale | 1.78 | 61 | U Utah | 0.53 |
| 12 | U Chicago | 1.75 | 62 | U Nebraska-Lincoln | 0.52 |
| 13 | U Minnesota Twin Cities | 1.7 | 63 | SUNY Stony Brook | 0.48 |
| 14 | UCLA | 1.68 | 64 | Boston U | 0.48 |
| 15 | Ohio State U | 1.64 | 65 | U Illinois Chicago | 0.46 |
| 16 | UPenn | 1.59 | 66 | Case Western Reserve | 0.46 |
| 17 | Princeton | 1.47 | 67 | U Cincinnati | 0.45 |
| 18 | UW | 1.45 | 68 | UConn | 0.45 |
| 19 | Purdue | 1.43 | 69 | Colorado State | 0.43 |
| 20 | Penn State | 1.43 | 70 | UC Irvine | 0.42 |
| 21 | UNC | 1.42 | 71 | Emory | 0.42 |
| 22 | Indiana University Bloomington | 1.34 | 72 | Syracuse | 0.4 |
| 23 | Michigan State | 1.34 | 73 | U Oklahoma | 0.39 |
| 24 | Northwestern | 1.17 | 74 | U South Carolina | 0.39 |
| 25 | Johns Hopkins | 1.16 | 75 | Washington State | 0.39 |
| 26 | U Florida | 1.14 | 76 | Oklahoma State | 0.37 |
| 27 | Texas A\&M | 1.12 | 77 | U Oregon | 0.36 |
| 28 | NYU | 1.08 | 78 | Texas Tech | 0.36 |
| 29 | U Maryland College Park | 1.05 | 79 | Oregon State | 0.35 |
| 30 | U Arizona | 0.98 | 80 | U New Mexico | 0.34 |
| 31 | Duke | 0.92 | 81 | UC San Francisco | 0.33 |
| 32 | U Georgia | 0.9 | 82 | Temple | 0.32 |
| 33 | U Pittsburgh | 0.89 | 83 | Kansas State | 0.32 |
| 34 | U Iowa | 0.89 | 84 | U Notre Dame | 0.32 |
| 35 | UC San Diego | 0.84 | 85 | Rice | 0.3 |
| 36 | U Southern California | 0.83 | 86 | U Houston | 0.29 |
| 37 | UC Davis | 0.82 | 87 | U Miami | 0.28 |
| 38 | U Virginia | 0.8 | 88 | Wayne State | 0.28 |
| 39 | Florida State | 0.75 | 89 | U South Florida | 0.28 |
| 40 | Virginia Tech | 0.74 | 90 | U Alabama | 0.28 |
| 41 | Rutgers - New Brunswick | 0.72 | 91 | U Alabama at Birmingham | 0.28 |
| 42 | Arizona State | 0.7 | 92 | Kent State | 0.27 |
| 43 | Caltech | 0.67 | 93 | Auburn | 0.27 |
| 44 | CU Boulder | 0.67 | 94 | Brandeis | 0.26 |
| 45 | U Rochester | 0.64 | 95 | U Delaware | 0.25 |
| 46 | U Missouri | 0.64 | 96 | George Washington | 0.24 |
| 47 | Iowa State | 0.64 | 97 | West Virginia | 0.24 |
| 48 | North Carolina State | 0.63 | 98 | UC Riverside | 0.23 |
| 49 | Washington U in St. Louis | 0.62 | 99 | Mississippi State | 0.23 |
| 50 | U Kansas | 0.62 | 100 | Tulane | 0.23 |

The 100 US universities that produced the most faculty across fields are shown in descending order, as well as the percent of all $n$=238,676 US-trained faculty produced by those universities. University names are compressed to save space using common abbreviations; see Data Availability for complete machine-readable data.

# Extended Data Table 4 | Prestige ranks, hiring, and placement, in US academia

| # | University | grads placed downward | grads placed upward | faculty hired from lower | faculty hired from higher | # | University | grads placed downward | grads placed upward | faculty hired from lower | faculty hired from higher |
|---|---|---|---|---|---|---|---|---|---|---|---|
| 1 | MIT | 4711 | 0 | 833 | 0 | 51 | U Virginia | 1416 | 336 | 409 | 1059 |
| 2 | Caltech | 1519 | 51 | 264 | 36 | 52 | U Southern California | 1197 | 398 | 300 | 1398 |
| 3 | Princeton | 3336 | 76 | 644 | 84 | 53 | Scripps Research (California) | 61 | 60 | 45 | 106 |
| 4 | Stanford | 5247 | 215 | 920 | 224 | 54 | U Pittsburgh | 1379 | 387 | 424 | 1061 |
| 5 | Harvard | 6260 | 464 | 1035 | 374 | 55 | Rensselaer Polytechnic Institute | 308 | 70 | 76 | 256 |
| 6 | UC Berkeley | 7168 | 447 | 780 | 579 | 56 | Michigan State | 2344 | 497 | 638 | 1421 |
| 7 | U Chicago | 3878 | 220 | 536 | 462 | 57 | Penn State | 2448 | 628 | 740 | 1652 |
| 8 | Yale | 3677 | 378 | 709 | 465 | 58 | UMass Amherst | 964 | 327 | 273 | 960 |
| 9 | Columbia | 3623 | 337 | 824 | 758 | 59 | Rutgers - New Brunswick | 1183 | 403 | 290 | 1383 |
| 10 | UPenn | 3330 | 277 | 741 | 699 | 60 | U Oregon | 657 | 167 | 122 | 692 |
| 11 | Cornell | 3889 | 314 | 877 | 687 | 61 | UC Irvine | 652 | 313 | 161 | 1057 |
| 12 | U Michigan | 5451 | 385 | 1150 | 971 | 62 | Clark | 96 | 26 | 25 | 84 |
| 13 | Carnegie Mellon | 1102 | 162 | 297 | 364 | 63 | U Arizona | 1599 | 437 | 458 | 1173 |
| 14 | UCLA | 3445 | 347 | 716 | 959 | 64 | UT Southwestern Med. Ctr. Dallas | 138 | 80 | 72 | 166 |
| 15 | U Wisconsin-Madison | 5066 | 375 | 1000 | 882 | 65 | Case Western Reserve | 654 | 268 | 237 | 568 |
| 16 | Johns Hopkins | 2168 | 346 | 563 | 624 | 66 | Emory | 668 | 246 | 190 | 806 |
| 17 | Northwestern | 2343 | 332 | 563 | 786 | 67 | Smith College | 15 | 2 | 6 | 12 |
| 18 | Brown | 1241 | 161 | 246 | 536 | 68 | Florida State | 1487 | 211 | 345 | 767 |
| 19 | Brandeis | 510 | 80 | 109 | 253 | 69 | Syracuse | 666 | 213 | 192 | 677 |
| 20 | NYU | 1912 | 472 | 493 | 1064 | 70 | Boston U | 663 | 320 | 233 | 1230 |
| 21 | U Illinois Urbana-Champaign | 4540 | 383 | 1017 | 1021 | 71 | University at Buffalo (SUNY) | 815 | 288 | 239 | 830 |
| 22 | U Rochester | 1215 | 193 | 321 | 437 | 72 | Albert Einstein College of Med. | 94 | 109 | 80 | 219 |
| 23 | UC San Francisco | 471 | 170 | 199 | 277 | 73 | Wesleyan | 41 | 15 | 8 | 84 |
| 24 | Duke | 1775 | 311 | 461 | 749 | 74 | U Notre Dame | 538 | 156 | 122 | 824 |
| 25 | UC San Diego | 1546 | 342 | 382 | 925 | 75 | U Illinois Chicago | 635 | 281 | 220 | 964 |
| 26 | Bryn Mawr | 104 | 13 | 19 | 55 | 76 | Tufts | 217 | 164 | 89 | 677 |
| 27 | SUNY Downstate Health Sci. | 67 | 18 | 21 | 27 | 77 | Vanderbilt | 796 | 360 | 318 | 1019 |
| 28 | UT Austin | 4099 | 368 | 769 | 1173 | 78 | Boston College | 311 | 117 | 62 | 624 |
| 29 | Washington U in St. Louis | 1190 | 212 | 260 | 619 | 79 | Catholic University of America | 185 | 48 | 48 | 187 |
| 30 | UC Santa Barbara | 1142 | 186 | 223 | 682 | 80 | Iowa State | 1024 | 310 | 373 | 904 |
| 31 | UW | 2475 | 550 | 662 | 1366 | 81 | U Georgia | 1587 | 387 | 588 | 1185 |
| 32 | U Minnesota Twin Cities | 3056 | 540 | 824 | 1203 | 82 | U Missouri | 1072 | 289 | 357 | 855 |
| 33 | Union Theological Seminary | 40 | 2 | 9 | 12 | 83 | U Kansas | 983 | 295 | 278 | 977 |
| 34 | Indiana Univ. Bloomington | 2760 | 284 | 612 | 884 | 84 | U Florida | 1712 | 637 | 618 | 1654 |
| 35 | UNC | 2576 | 502 | 732 | 1141 | 85 | U Denver | 191 | 45 | 40 | 212 |
| 36 | SUNY Stony Brook | 875 | 201 | 246 | 579 | 86 | Thomas Jefferson | 45 | 53 | 50 | 88 |
| 37 | Alfred | 30 | 16 | 14 | 10 | 87 | Cold Spring Harbor | 3 | 12 | 3 | 38 |
| 38 | Princeton Theol. Seminary | 46 | 11 | 11 | 28 | 88 | Georgetown | 255 | 149 | 78 | 672 |
| 39 | U Iowa | 1723 | 234 | 422 | 665 | 89 | New York Medical College | 30 | 26 | 15 | 63 |
| 40 | Rice | 576 | 108 | 134 | 439 | 90 | UC Riverside | 333 | 194 | 89 | 765 |
| 41 | U Maryland College Park | 1987 | 358 | 534 | 1197 | 91 | Virginia Tech | 1178 | 396 | 345 | 1198 |
| 42 | Georgia Tech | 1066 | 212 | 265 | 740 | 92 | DePaul | 38 | 16 | 12 | 40 |
| 43 | Jewish Theological Seminary | 4 | 10 | 2 | 23 | 93 | SUNY Upstate Medical | 24 | 33 | 19 | 66 |
| 44 | New School | 116 | 23 | 32 | 90 | 94 | U Utah | 780 | 288 | 198 | 1144 |
| 45 | Purdue | 2718 | 368 | 793 | 1097 | 95 | Teachers College Columbia | 58 | 42 | 18 | 153 |
| 46 | UC Davis | 1409 | 333 | 433 | 1130 | 96 | U Maryland Baltimore | 122 | 71 | 59 | 152 |
| 47 | UC Santa Cruz | 337 | 152 | 104 | 524 | 97 | Texas A\&M | 1846 | 530 | 566 | 1704 |
| 48 | Meharry Medical College | 30 | 3 | 7 | 3 | 98 | Rush | 48 | 28 | 27 | 57 |
| 49 | CU Boulder | 1264 | 244 | 303 | 976 | 99 | Claremont Graduate | 139 | 53 | 25 | 292 |
| 50 | Ohio State U | 3066 | 507 | 893 | 1485 | 100 | UConn | 644 | 319 | 191 | 1157 |

The 100 most prestigious universities, as inferred from patterns in faculty hiring ($n$=238,281 total faculty; see Methods) and shown in descending order. Columns shown the number of graduates (grads) of each university who become faculty (are placed, i.e. network out-degrees) at lower/higher prestige universities, and the number of faculty employed by each university (i.e. network in-degrees) who earned their degree from a lower/higher prestige university. University names are compressed to save space using common abbreviations; see Data Availability for complete machine-readable data.

**Extended Data Table 5 | Comparison of empirical prestige hierarchies with network null model**

| Field | # of null model draws less hierarchical than empirical (out of 1000) |
|---|---|
| Pharmacy | 880 |
| Immunology | 763 |
| Pathology | 719 |
| Agronomy | 666 |
| Horticulture | 561 |
| Natural Resources | 540 |
| Entomology | 399 |
| Anatomy | 220 |
| Near and Middle Eastern Languages and Cultures | 152 |
| Pharmacology | 124 |
| Plant Pathology | 124 |
| Evolutionary Biology | 92 |
| Forestry and Forest Resources | 58 |
| Biomedical Engineering | 42 |
| Comparative Literature | 34 |
| Biophysics | 26 |
| Ecology | 21 |
| Veterinary Medical Sciences | 20 |
| Germanic Languages and Literatures | 16 |
| Environmental Health Sciences | 16 |
| Animal Sciences | 10 |
| Communication Disorders and Sciences | 9 |
| Microbiology | 8 |
| Asian Studies | 8 |
| Counselor Education | 6 |
| Nutrition Sciences | 5 |
| Health, Physical Education, Recreation | 4 |
| Public Health | 4 |
| Operations Research | 3 |
| Environmental Sciences | 2 |
| Exercise Science, Kinesiology, Rehab, Health | 1 |
| Soil Science | 1 |
| Genetics | 1 |

Hierarchies encoded in academia and in each field and domain were compared independently with a set of 1,000 hierarchies generated using a degree-preserving null model (Methods). A column shows the number of null model draws (from a total possible 1,000) that were more hierarchical than the empirical network, as measured by the fraction of edges in each network aligned with the direction of the hierarchy. Empirical hierarchies that were more hierarchical than all 1,000 null model hierarchies are omitted.

# Extended Data Table 6 | Rank change by domain and field

| Domain / Field | moved upward | moved downward | self-hired | avg. mvmt. upward | avg. mvmt. downward |
|---|---|---|---|---|---|
| *Academia* | 18% | 71% (-4%) | 11% (+4%) | 14% (+1%) | 28% (+1%) |
| *Applied Sciences* | 17% | 73% | 9% | 12% | 29% |
| Accounting | 9% | 85% | 6% | 9% | 37% |
| Agronomy | 21% | 50% | 29% | 26% | 39% |
| Animal Sciences | 23% | 57% | 20% | 21% | 34% |
| Architecture | 7% | 78% | 15% | 10% | 39% |
| Business Administration | 10% | 75% | 14% | 9% | 29% |
| Finance | 6% | 90% | 4% | 11% | 38% |
| Food Science | 14% | 62% (-16%) | 24% (+15%) | 20% | 36% |
| Horticulture | 20% | 53% | 26% | 25% | 40% |
| Management | 11% | 81% | 8% | 12% | 34% |
| Management Information Sys. | 8% | 79% | 13% | 12% | 36% |
| Marketing | 9% | 87% | 4% | 9% | 38% |
| Plant Sciences | 13% | 59% | 28% | 20% | 37% |
| Soil Science | 16% | 56% | 27% | 25% | 36% |
| Urban & Regional Planning | 10% | 75% | 15% | 14% | 35% |
| *Education* | 14% | 70% (-4%) | 15% (+5%) | 15% | 35% |
| Counselor Education | 14% | 59% | 27% (+9%) | 17% | 36% |
| Curriculum & Instruction | 13% | 67% (-8%) | 20% (+9%) | 15% | 38% |
| Education | 11% | 60% | 29% (+7%) | 15% | 35% |
| Education Administration | 11% | 70% | 19% (+6%) | 15% | 38% |
| Special Education | 9% | 57% (-15%) | 35% (+14%) | 19% | 39% |
| *Engineering* | 14% (+2%) | 73% | 13% | 12% | 29% (-2%) |
| Aerospace Engineering | 10% | 71% | 19% | 13% | 37% |
| Agricultural Engineering | 17% | 66% | 17% | 19% | 34% |
| Civil Engineering | 12% | 77% | 12% | 11% | 33% |
| Electrical Engineering | 12% | 74% | 14% | 11% | 28% |
| Environmental Engineering | 12% | 74% | 13% | 12% | 34% |
| Industrial Engineering | 11% | 76% | 13% | 14% | 35% |
| Materials Engineering | 11% | 75% | 14% | 13% | 35% |
| Mechanical Engineering | 11% | 78% | 11% | 12% | 34% |
| Operations Research | 13% | 73% | 14% | 13% | 36% |
| Systems Engineering | 8% | 66% | 26% | 16% | 35% |
| *Humanities* | 12% (+1%) | 82% | 6% | 10% | 31% |
| Art History & Criticism | 7% | 88% | 5% | 8% | 39% |
| Asian Languages | 11% | 80% | 9% | 12% | 39% |
| Asian Studies | 14% | 78% | 8% | 11% | 37% |
| Classics & Classical Languages | 5% | 92% | 3% | 8% | 46% |
| Comparative Literature | 9% | 84% | 7% | 16% | 44% |
| English Language & Literature | 10% | 86% | 4% (+2%) | 12% | 36% |
| French Language & Literature | 8% | 86% | 6% | 12% | 39% |
| Germanic Languages & Lit. | 12% | 84% | 4% | 11% | 40% |
| History | 8% | 89% | 4% | 7% | 35% |
| Linguistics | 12% | 78% | 10% | 14% | 38% |
| Music | 8% | 85% | 7% | 9% | 38% |
| Near/Mid. Eastern Lang./Cultures | 17% | 72% | 12% | 12% | 38% |
| Philosophy | 8% | 88% | 4% | 8% | 36% |
| Religious Studies | 5% | 80% | 15% | 13% | 41% |
| Slavic Languages & Literatures | 7% | 88% | 6% | 11% | 44% |
| Spanish Language & Literature | 11% | 82% | 7% | 13% | 38% |
| Theatre Literature, History & Crit. | 7% | 78% | 15% | 12% | 38% |
| Theological Studies | 11% | 53% | 36% (-9%) | 14% | 31% |
| *Math & Computing* | 13% (+2%) | 79% (-4%) | 8% (+1%) | 11% (+1%) | 28% |
| Computer Engineering | 13% | 71% | 16% | 12% | 32% |
| Computer Science | 12% | 80% | 8% | 9% | 29% |
| Information Science | 14% | 69% | 17% | 12% | 34% |
| Information Technology | 7% | 70% | 23% | 15% | 38% |
| Mathematics | 9% | 87% (-5%) | 4% (+4%) | 10% | 33% |
| Statistics | 7% | 86% | 7% (+4%) | 10% | 36% |

| Domain / Field | moved upward | moved downward | self-hired | avg. mvmt. upward | avg. mvmt. downward |
|---|---|---|---|---|---|
| *Medicine & Health* | 21% (-3%) | 57% (-5%) | 22% (+8%) | 19% | 32% (+1%) |
| Communication Disord.& Sci. | 10% | 64% | 25% | 14% | 40% |
| Environmental Health Sci. | 8% | 50% | 42% | 21% | 40% |
| Epidemiology | 12% | 56% (-9%) | 32% (+9%) | 24% | 43% |
| Exercise Sci., Kines., Rehab | 16% | 58% (-8%) | 26% (+11%) | 19% | 36% |
| Genetics | 14% | 59% | 27% | 20% | 36% |
| Health, Phys. Ed., Recreation | 9% | 67% | 24% | 14% | 41% |
| Human Dev. & Family Sci. | 13% | 66% | 21% | 18% | 36% |
| Immunology | 18% | 60% | 22% (+8%) | 20% | 42% |
| Nursing | 14% (-6%) | 52% | 34% (+6%) | 19% | 36% |
| Nutrition Sciences | 16% | 59% | 25% (+9%) | 17% | 36% |
| Pharmaceutical Sciences | 11% | 53% | 36% (+11%) | 19% | 36% |
| Pharmacology | 20% | 59% (-10%) | 21% (+9%) | 21% | 37% |
| Pharmacy | 13% | 42% | 45% (+10%) | 20% | 40% |
| Physiology | 16% | 63% | 21% (+9%) | 18% | 36% |
| Public Health | 15% | 58% (-7%) | 27% (+7%) | 19% | 38% |
| Social Work | 14% | 71% | 14% | 13% | 33% |
| Speech & Hearing Sciences | 10% | 65% | 25% | 12% | 36% |
| Veterinary Medical Sciences | 20% | 45% | 36% (+7%) | 22% | 32% |
| *Natural Sciences* | 20% (+1%) | 69% (-3%) | 11% (+2%) | 15% (+1%) | 28% |
| Anatomy | 10% | 56% | 34% | 20% | 43% |
| Astronomy | 9% | 82% | 10% | 12% | 36% |
| Atmosph.Sci. & Meteorology | 12% | 70% | 18% | 14% | 33% |
| Biochemistry | 15% | 75% | 10% | 16% | 36% |
| Biological Sciences | 11% | 80% | 8% | 14% | 36% |
| Biomedical Engineering | 16% | 62% | 22% | 16% | 38% |
| Biophysics | 11% | 55% | 34% | 21% | 39% |
| Biostatistics | 11% | 65% | 24% | 15% | 39% |
| Cell Biology | 17% | 67% | 16% | 18% | 36% |
| Chemical Engineering | 11% | 81% | 8% | 11% | 34% |
| Chemistry | 11% | 83% | 6% (+3%) | 11% | 33% |
| Ecology | 16% | 67% | 16% | 19% | 38% |
| Entomology | 21% | 61% | 18% | 22% | 39% |
| Environmental Sciences | 12% | 68% | 20% | 15% | 38% |
| Evolutionary Biology | 16% | 71% | 14% | 14% | 38% |
| Forestry & Forest Resources | 17% | 51% | 33% | 23% | 36% |
| Geology | 12% | 80% | 9% | 12% | 33% |
| Marine Sciences | 13% | 62% | 25% | 18% | 35% |
| Microbiology | 17% | 66% (-8%) | 17% (+8%) | 20% | 40% |
| Molecular Biology | 12% | 74% | 14% | 16% | 37% |
| Natural Resources | 14% | 54% | 32% | 20% | 39% |
| Neuroscience | 16% | 68% | 17% (+6%) | 18% | 36% |
| Pathology | 23% | 52% | 25% (+5%) | 26% | 39% |
| Physics | 10% | 81% | 8% | 9% | 31% |
| Plant Pathology | 19% | 62% | 19% | 23% | 40% |
| *Social Sciences* | 14% | 79% (-2%) | 7% (+1%) | 12% | 29% |
| Agricultural Economics | 13% | 69% | 18% | 14% | 36% |
| Anthropology | 11% | 84% | 5% | 10% | 36% |
| Crim. Justice & Criminology | 8% | 80% | 11% | 11% | 36% |
| Economics | 6% | 91% | 3% | 6% | 31% |
| Educational Psychology | 13% | 64% | 23% | 18% | 39% |
| Gender Studies | 8% | 79% | 12% | 15% | 37% |
| Geography | 12% | 79% | 9% | 14% | 37% |
| International Affairs | 7% | 75% | 17% | 17% | 42% |
| Political Science | 10% | 86% | 4% | 6% | 32% |
| Psychology | 15% | 77% | 8% | 13% | 32% |
| Sociology | 9% | 86% | 5% | 9% | 34% |

Faculty movements within the prestige hierarchies are shown for academia (blue, bold italics), domains (highlighted, bold italics) and fields (e.g. Physics, or Nutritional Sciences); total $n = 238,281$. Three columns show how faculty flows break down by movement up the hierarchy, movement down the hierarchy, or self-hiring. In instances in which rates vary significantly by gender (two-sided $z$-test for proportions, Benjamini-Hochberg corrected, $p < 0.05$), values in parentheses show the difference in rates between women vs men, such that positive values indicate larger percentages for women. Two columns show the average movement distance (avg. mvmt.) upward for those moving up, and downward for those moving down. In instances in which distances vary significantly by gender (K.S. test for difference in distance distributions; Benjamini-Hochberg corrected $p < 0.05$), values in parentheses show the difference in distances between women versus men, such that positive values indicate larger movements for women. Statistical tests for differences by gender apply to only those $n = 204,330$ faculty with gender annotations (Methods).

# Reporting Summary

## Statistics

For all statistical analyses, confirm that the following items are present in the figure legend, table legend, main text, or Methods section.

| n/a | Confirmed | |
|---|---|---|
| ☐ | ☒ | The exact sample size (*n*) for each experimental group/condition, given as a discrete number and unit of measurement |
| ☒ | ☐ | A statement on whether measurements were taken from distinct samples or whether the same sample was measured repeatedly |
| ☐ | ☒ | The statistical test(s) used AND whether they are one- or two-sided<br>*Only common tests should be described solely by name; describe more complex techniques in the Methods section.* |
| ☐ | ☒ | A description of all covariates tested |
| ☐ | ☒ | A description of any assumptions or corrections, such as tests of normality and adjustment for multiple comparisons |
| ☐ | ☒ | A full description of the statistical parameters including central tendency (e.g. means) or other basic estimates (e.g. regression coefficient) AND variation (e.g. standard deviation) or associated estimates of uncertainty (e.g. confidence intervals) |
| ☐ | ☒ | For null hypothesis testing, the test statistic (e.g. *F*, *t*, *r*) with confidence intervals, effect sizes, degrees of freedom and *P* value noted<br>*Give P values as exact values whenever suitable.* |
| ☒ | ☐ | For Bayesian analysis, information on the choice of priors and Markov chain Monte Carlo settings |
| ☒ | ☐ | For hierarchical and complex designs, identification of the appropriate level for tests and full reporting of outcomes |
| ☐ | ☒ | Estimates of effect sizes (e.g. Cohen's *d*, Pearson's *r*), indicating how they were calculated |

*Our web collection on statistics for biologists contains articles on many of the points above.*

## Software and code

Policy information about availability of computer code

| | |
|---|---|
| Data collection | See Data and Approach for information on original data obtained from AARC. See Methods for information on additional information on how universities were hand-annotated with country labels. |
| Data analysis | Open-source code related to this study is available at https://doi.org/10.5281/zenodo.6941612. |

For manuscripts utilizing custom algorithms or software that are central to the research but not yet described in published literature, software must be made available to editors and reviewers. We strongly encourage code deposition in a community repository (e.g. GitHub). See the Nature Portfolio guidelines for submitting code & software for further information.

## Data

Policy information about availability of data

All manuscripts must include a data availability statement. This statement should provide the following information, where applicable:
- Accession codes, unique identifiers, or web links for publicly available datasets
- A description of any restrictions on data availability
- For clinical datasets or third party data, please ensure that the statement adheres to our policy

All network data associated with this study, and all data contained in Extended Data tables are freely available in machine-readable format at https://doi.org/10.5281/zenodo.6941651. Explorable visualizations of faculty hiring networks are available at larremorelab.github.io/us-faculty/, and for university ranks, at larremorelab.github.io/us-institutions/.

# Human research participants

Policy information about <u>studies involving human research participants and Sex and Gender in Research.</u>

| | |
|---|---|
| Reporting on sex and gender | This manuscript discusses gender, but not sex. As stated in Methods, we used self-identified genders when possible, and machine-annotated genders otherwise:<br><br>Self-identified gender annotations were provided for 6% of faculty in the unprocessed dataset. In order to annotate the remaining faculty with gender estimates, we used a two-step process based on first and last name. First, complete names were passed to two offline dictionaries: a hand-annotated list of faculty employed at Business, Computer Science, and History departments (corresponding to data used in Ref. [27]), and the open-source python package gender-guesser [58]. Both dictionaries responded with one of the following classifications: female, male, or unable to classify. Second, for the cases where the dictionaries either disagreed or agreed but were unable to assign a gender to the name, we queried Ethnea [59] and used the gender they assigned the name (if any). Using this approach we were able to annotate 85% of faculty with man or woman labels. Faculty whose names could not be associated with a gender were excluded from analyses of gender but included in other analyses. This methodology associates names with binary (man/woman) labels because of technical limitations inherent to name-based gendering methodologies, but we recognize that gender is nonbinary. The use of these binary gender labels is not intended to reinforce the gender binary. |
| Population characteristics | Our analysis examines tenured or tenure-track faculty employed in the years spanning 2011 and 2020 at 368 PhD-granting universities in the U.S., each of whom is annotated by their doctoral institution, year of doctorate, faculty rank, and faculty gender. To be included in our analysis, a professor must be a member of the tenured or tenure-track faculty at a department that appears in the majority of sampled years, which yields n = 295, 089 faculty in 10,612 departments. |
| Recruitment | This dataset resulted from cleaning and preprocessing a larger U.S. faculty census obtained under a Data Use Agreement with the Academic Analytics Research Center (AARC). This dataset spanned all tenure-track and tenured faculty at U.S. PhD-granting institutions, between 2011-2020. |
| Ethics oversight | After consultation with the University of Colorado Boulder IRB, protocol submission and approval was deemed unnecessary for the present study, due to is secondary use of publicly available data. |

Note that full information on the approval of the study protocol must also be provided in the manuscript.

# Field-specific reporting

Please select the one below that is the best fit for your research. If you are not sure, read the appropriate sections before making your selection.

☐ Life sciences  ☒ Behavioural & social sciences  ☐ Ecological, evolutionary & environmental sciences

For a reference copy of the document with all sections, see <u>nature.com/documents/nr-reporting-summary-flat.pdf</u>

# Behavioural & social sciences study design

All studies must disclose on these points even when the disclosure is negative.

| | |
|---|---|
| Study description | This study quantitatively analyzes patterns found in qualitative data, namely the records of individual tenure-track or tenured faculty at U.S. PhD-granting institutions between 2011-2020. Namely, we analyze PhD institution, current department and institution, faculty rank, and gender. By observing new entrants to the dataset over time, or departures over time, we also analyze hiring and attrition of said U.S. tenured or tenure-track faculty. |
| Research sample | All tenure-track and tenured faculty at U.S. PhD-granting institutions, except for faculty of Law and Medical schools. This sample represents a ten-year annual census of these faculty and is representative due to its complete coverage; this is not a random subsample of the population being studied. |
| Sampling strategy | Census sampling was used by the original data providers (AARC). In some cases, data were reported to the AARC directly by institutions themselves. In all other cases, the AARC (or their affiliates) collected faculty rosters and doctoral degree information from public-facing university webpages, annually. The data provided to the research team spanned only 2011-2020, though the sampling strategy has been in use by the AARC or their affiliates for years prior to our sample frame. |
| Data collection | Our data resulted from cleaning and preprocessing the larger academic census dataset obtained under a Data Use Agreement with the Academic Analytics Research Center (AARC), who collected the original dataset as described above. Please see Methods for detailed descriptions of the nine key cleaning steps, and two key annotation steps that were used prior to the manuscript's analyses. |
| Timing | 2011 to 2020 |
| Data exclusions | A complete description of data exclusions and cleaning — and which data were excluded or included for each analysis — is included in Methods. |

| Non-participation | Not applicable. |
|---|---|
| Randomization | Not applicable. |

# Reporting for specific materials, systems and methods

We require information from authors about some types of materials, experimental systems and methods used in many studies. Here, indicate whether each material, system or method listed is relevant to your study. If you are not sure if a list item applies to your research, read the appropriate section before selecting a response.

## Materials & experimental systems

| n/a | Involved in the study |
|---|---|
| ☒ ☐ | Antibodies |
| ☒ ☐ | Eukaryotic cell lines |
| ☒ ☐ | Palaeontology and archaeology |
| ☒ ☐ | Animals and other organisms |
| ☒ ☐ | Clinical data |
| ☒ ☐ | Dual use research of concern |

## Methods

| n/a | Involved in the study |
|---|---|
| ☒ ☐ | ChIP-seq |
| ☒ ☐ | Flow cytometry |
| ☒ ☐ | MRI-based neuroimaging |

