## [Peer Review File · Nature]

Manuscript Title: Quantifying hierarchy and dynamics in U.S. faculty hiring and retention

Reviewer Comments & Author Rebuttals

Reviewer Reports on the Initial Version:

Referee #1 (Remarks to the Author):

This is an important paper with strong empirical evidence and implications across all fields of science. There are, however, several improvements that can be made, largely to enhance the accessibility of the work for a broader audience. Comments are made in no particular order. The conceptualization and operationalization of the “U.S. faculty hiring network” needs to be made explicit much earlier in the manuscript. At present, the Introduction refers to “the complete U.S. faculty hiring” and then ends with a note stating that the approach “enables analyses of hires...from outside the U.S.” The first paragraph implies that faculty must be tenured or tenure-track at one of 371 universities in the US and that the hiring network is constructed from this sampling frame; however, it then suggests that the requirements are “relaxed” for other analyses. The Results are then presented with an interplay between the within-set and expanded set. I would strongly urge the authors to clarify this in the Introduction and Data and to structure the Results to focus first on the within-set and then the more expansive data set (making it explicit in the Results as well). The paragraph on “top-10” departments is unclear. I have inferred that the 810 top-10 departments are those 10 departments within each field that have the highest number of graduates employed within the dataset. However, this could be clarified. The language is difficult to parse at present. Furthermore, one may want to identify some departments or institutions beyond the top five. As a minor aside, every institution will want to see where it falls in the analysis. The supplemental data provide robust information on field, but fail to provide any institutional data. I would recommend that the authors provide at least those institutions referenced in the text (i.e., “top-10” “top-5”), but ideally a list of the top 77 institutions, which are delineated in Fig.1.A. Related to institutions, it is unclear throughout the narrative whether the main unit of analysis is domain or institution. The abstract implies that this paper is about institutional structure; however, all of the figures are focused on disciplinary differences. This is a critical point for the organization of the paper. At present, there is no clear research question. The goal for the paper, however, is implied in this statement: “Without a complete understanding of how prestige and hiring manifest across different fields, our understanding of the underlying causes of prestige hierarchies and their ultimate effects on the academic system, positive or negative, remain limited, inhibiting our ability

to develop effective policies aimed at accelerating scientific discovery or increasing the diversity of the professoriate.” The lack of a specific charge is also communicated in the organization of the Results, which moves from one analysis to the next, without a clear guiding theory. The data presented in this paper is of high quality and importance; therefore, I would strongly urge the authors to reformat the paper with greater clarity to make the work more accessible.

Related to this, the authors may want to describe the unit of analysis in regards to individuals. For example, the double counting of disciplinary affiliations suggests that the individual is not the unit of analysis, but rather the individual-discipline pair. This needs greater explanation and justification. The text and figures on up/down mobility could use greater clarification. For example, the text states that, for the typical professor who moves down in the prestige hierarchy, they fall 46.3% downward in Classics & Classical Languages. However, Table S2 suggests that professors in Classics & Classical Languages have downward movement at 41%, on average. Is 46% the high end of that range rather than the average? This should be clarified in the text and Table S2. Furthermore, the statement “while the typical professor who moves upward, of whom there are very few, ascends between 6.5% (agronomy) and 26.1% (Classics and Classical Languages) of the hierarchy. This is also, confusing. It is unclear whether this number should represent the proportion of those who move upwards or the rank difference traversed. Neither, however, match Table S2, which suggests that only 4% of Classics & Classical Literature professors have upward mobility and, when they do, they tend to only have a 10% difference in rank. Clarification on these differences would be helpful.

It would be useful to bring in more contextual information to understand the results. For example, might some of the explanation in Medicine & Health be due to the context of university hospitals and practices of soft-money hiring for assistant professors? Could the dominance of Nursing in the Medicine & Health category be an explanation for the gender differences observed in this field? Prestige appears to be the central concept in this text. However, I cannot find a single definition of prestige or a description in how it was measured in this analysis. I would encourage the authors to include this explicitly in the Data section.

Academic rank is introduced in the last paragraph of Results. However, there is an important difference noted for Assistant Professors that bears greater examination and discussion. As the authors note, faculty production inequality is universally lower for junior faculty. That is, there are a wider array of institutions producing junior faculty. This is essential for the narrative, which is about hiring networks. In effect, what we are seeing here is that the most pronounced inequity is not in hiring, but in promotion and tenure. The higher concentration of affiliations at the senior level suggests that attrition diminishes institutional diversity. One might argue that this entire analysis should only be at the Assistant level, if focused on direct hiring links. At present, the obscuring of the hiring pathway is incredibly problematic—as the authors acknowledge, we do not know whether self-hires are returnees or whether those with upward mobility traversed across several institutions first. The arguments around “academic inbreeding” do not hold if there was high mobility before returning to the home unit.

At present, most of the analyses on gender and rank demonstrate that women make up a larger proportion of the Assistant rank. This is well observed. What is not evident, is how much hiring and institutional prestige factor in the attrition of women scholars. The authors should remove the focus on composition and interrogate the institutional factors as relevant to gender at the Assistant level as compared to more senior levels.

The paragraph that suggests a zero-sum game around diversity policies lacks clarity. First, diversity should be clearly defined—is this institutional diversity, field permeability and interdisciplinarity, or

socio-demographic diversity. Furthermore, the example is unclear, how is increasing representation of women in Engineering not an individual field-level effort? The argument here is important and unclear.

There is relatively little analysis and discussion of non-US PhDs. What is presented is entirely at the level of field, and not institution. Once one controls by field, how does institutional prestige interact with the hiring of out-network PhDs? Furthermore, although the data may not be present to interrogate this, the authors should at least acknowledge the large influx of foreign-born scholars who are educated at the baccalaureate levels and receive PhD training here. This is relevant for the hiring network, given research that suggests foreign-born scholars may exhibit different characteristics in salary and selection in the academic job market.

The second paragraph is unnecessarily technical. These comments can and should be expressed as conceptualizations in the introduction, to be operationalized in the methods section.

How were the 391 “degree-producing U.S. institutions” identified—do the authors mean all doctoral-degree granting institutions? If so, this should be specified.

The authors define gender as “male/female”, though these terms are usually used for sex; furthermore, they should acknowledge the expansive gender identities beyond these dominant two. How do the authors determine fields that were “too small for analysis”? 4595 faculty seems substantial and exceeds the 1k faculty limits mentioned in the supplemental documents. To avoid exclusion, why not aggregate at the higher domain levels that are presented in the text and figures? Why exclude at the specialty level? This makes fields like social sciences highly underrepresented. Several of the figures could be improved for readability. Figure 1.C should be reoriented to avoid having to read sideways. “Academia” should be removed from Figure 2—if the total should be represented, perhaps with a dashed line? Figure 3 should be reoriented to avoid sideways reading. Be explicit about what is meant by “top five” in Figure 6. Consider making Figure 7 vertical to improve readability.

I would recommend a few small editorial changes. For example: “academic and research ecosystem” \diamond “academic and research ecosystems”; “2013and 2017” \diamond “2013 and 2017”; avoid double “most” in “most fields tend...most fields employ”; include the term or symbol for percentages on the last sentence of page 13.

The first line could be strengthened. I would argue that faculty hiring is not indirectly related to teaching, training, and mentoring—these are very directly related. Furthermore, there is a noticeable omission of access and networks here; there is a strong relationship between hiring and access to certain networks and opportunities. I would recommend a stronger and more precise opening statement.

The authors have greatly expanded their data and analysis beyond their 2015 Science Advances paper. However, they may want to provide some specific language detailing the difference in both approach and findings between these two papers. Given the similarity in title and abstract, it may not be clear what the fundamental contribution of the second paper is, besides increased sample size.

Referee #2 (Remarks to the Author):

Dear authors,

This is an interesting and impressive paper, which uses a large data set to analyse faculty hiring in the U.S. Moreover, the approach used appears as well chosen and, to the extent that I am able to judge, the analysis has been done in a methodologically sound manner. Moreover, the presentation is clear, the figures are adequate, and the paper is well-structured. The main finding is that U.S. academia – across all fields – is organised in a core-periphery structure where a minority of elite institutions produces a majority of all faculty. The study also finds that one in nine professors are hired by their own doctoral university, which is seen as an indication of “self-hiring” being common. Self-hires are also more prevalent when women are appointed, and among higher-ranked institutions. In my view these findings are important, yet at the same time, they are not really new. The hierarchy of U.S. academia, both generally and in specific disciplines, has been studied before, as documented in the reference list. Especially, two studies involving authors of the current manuscript (ref 22 and ref 28), reach similar conclusions. While the data set used 191,000 is considerably higher, the main findings echo the Science advances paper published in 2015. Hence, the study under review here confirms previous research rather than revealing new unexpected patterns.

Moreover, I find that the paper would make an even stronger impression if the empirical findings could be contextualized and compared more systematically. For example, is it possible to find changes over time? Is inequality rising or diminishing in science? A recent paper (<https://www.pnas.org/content/118/7/e2012208118> found that inequality in terms of citations is increasing, could the same observation apply to academic recruitment? Furthermore, the unequal distribution of resources and rewards in science has been studied for a long time (e.g. Merton) and the distribution found here (that 20 % of the institutions contribute to 80% of the faculty produced) has been seen found also in terms of the production of papers (20% of the authors produce 80% of all scientific papers). Hence, it might be useful to discuss your very interesting findings in relation to how the Pareto principle can be applied to science more generally.

Another way of discussing these results would be to reflect on what they say about the structure of U.S. Science. Or more concretely, do these findings reflect a more general structure in science or are it due to specific patterns in an American context. In my experience would, for example, the levels of self-hiring found here to be seen as very low in a European context.

Overall, I find that contextual comparisons would open for conclusions that more directly could engage, and contribute to, current debates about inequality, discrimination of certain groups, etc. in academia. This would make the paper even more relevant to a large group of researchers and policymakers interested in how science is structured and organised.

Author Rebuttals to Initial Comments:

Referee 1

This is an important paper with strong empirical evidence and implications across all fields of science. There are, however, several improvements that can be made, largely to enhance the accessibility of the work for a broader audience. Comments are made in no particular order.

We thank the referee for these encouraging and constructive comments.

The conceptualization and operationalization of the “U.S. faculty hiring network” needs to be made explicit much earlier in the manuscript. At present, the Introduction refers to “the complete U.S. faculty hiring” and then ends with a note stating that the approach “enables analyses of hires...from outside the U.S.” The first paragraph implies that faculty must be tenured or tenure-track at one of 371 universities in the US and that the hiring network is constructed from this sampling frame; however, it then suggests that the requirements are “relaxed” for other analyses. The Results are then presented with an interplay between the within-set and expanded set. I would strongly urge the authors to clarify this in the Introduction and Data and to structure the Results to focus first on the within-set and then the more expansive data set (making it explicit in the Results as well).

These suggestions have led us to a number of changes, which we think improve the flow and scope of the paper. All changes are guided by our belief—clarified by the reviewer’s comment—that our analysis of U.S. faculty and faculty hiring *should* include faculty trained outside the U.S. whenever possible. To that end, we have made the following concrete changes:

- Introduction & Data sections emphasize our goal of this broad frame, whenever possible.
- New Fig. 1A leads the Results by explicitly accounting for non-U.S. doctorates (as well as faculty without doctorates).
- All “percentage of all faculty” statistics, which were previously computed using only U.S.-trained faculty in the denominator, are now computed using *all* faculty in the denominator. For instance, a 10% self-hire rate now implies that 10% of *all* U.S. faculty are self-hires, rather than 10% of U.S.-trained faculty.
- New Fig. 1B analyzes the origins of non-U.S. trained faculty by domain of study.
- New Fig. 1C shows that faculty trained in Canada and the U.K. show no statistical difference in rates of attrition than U.S.-trained faculty.
- However, New Fig. 1D shows faculty trained outside the U.S., Canada, and the U.K. exhibit markedly higher rates of attrition.

The paragraph on “top-10” departments is unclear. I have inferred that the 810 top-10 departments are those 10 departments within each field that have the highest number of graduates employed within the dataset. However, this could be clarified. The language is difficult to parse at present. Furthermore, one may want to identify some departments or institutions beyond the top five.

We have clarified this paragraph's language in two ways. First, we now introduce the idea that among 107¹ fields, there are 1070 potential top-10 spots. Second, we clarify that these top-10s are actually by the field level measures of prestige, derived from hiring networks, and not top-10s by number of faculty produced.

As a minor aside, every institution will want to see where it falls in the analysis. The supplemental data provide robust information on field, but fail to provide any institutional data. I would recommend that the authors provide at least those institutions referenced in the text (i.e., "top-10" "top-5"), but ideally a list of the top 77 institutions, which are delineated in Fig.1.A.

This is a great point. The very first thing that many readers will do will be to look up their current or past institutions. To make this more streamlined, and also more engaging, we've now added two things:

- Two new supplementary tables listing the top-100 institutions by prestige and by faculty production.
- A new interactive data visualization in which users can select their institution, and see where it is placed based on either faculty production or prestige, across fields and domains. The visualization allows people to show two institutions at once, so they can compare — another thing we imagine readers will probably want to do! The visualization can be found at larremorelab.github.io/us-institutions.

Related to institutions, it is unclear throughout the narrative whether the main unit of analysis is domain or institution. The abstract implies that this paper is about institutional structure; however, all of the figures are focused on disciplinary differences. This is a critical point for the organization of the paper. At present, there is no clear research question. The goal for the paper, however, is implied in this statement: "Without a complete understanding of how prestige and hiring manifest across different fields, our understanding of the underlying causes of prestige hierarchies and their ultimate effects on the academic system, positive or negative, remain limited, inhibiting our ability to develop effective policies aimed at accelerating scientific discovery or increasing the diversity of the professoriate." The lack of a specific charge is also communicated in the organization of the Results, which moves from one analysis to the next, without a clear guiding theory. The data presented in this paper is of high quality and importance; therefore, I would strongly urge the authors to reformat the paper with greater clarity to make the work more accessible.

This comment has led us to reflect on the framing of the paper, particularly in light of other suggestions about scope, and our ability now to make longitudinal observations. Primarily, we believe that the composition of the professoriate (driven by the structure of the hiring market and non-random patterns of attrition) is of fundamental importance to understanding (1) all the things

¹ An important consequence of our negotiation for new/expanded data from the AARC is that we can now include numerous additional fields. We have also slightly adjusted our inclusion criteria to be more inclusive. Together, these changes raise the total field count from 81 to 107. We note that individuals in these fields *were previously, and still are*, included in all domain-level and academia-level analyses.

that faculty do, including teaching, research, and policy advising; and (2) the ability of institutions and funders to address inequalities, be they of prestige, national origin, or gender. We hope that this is now more clearly communicated in the Introduction and Discussion, especially.

Related to this, the authors may want to describe the unit of analysis in regards to individuals. For example, the double counting of disciplinary affiliations suggests that the individual is not the unit of analysis, but rather the individual-discipline pair. This needs greater explanation and justification.

We guide our unit of analysis choice by considering what one would expect if one were to ask a question such as, "What fraction of professors in Physics were trained in Europe?" Here, we believe that one would expect a professor of Physics & Astronomy to be included in that calculation. However, we would also expect that same person to be included when asking, "What fraction of professors in Astronomy were trained in Europe?" On the other hand, if one were to ask "What fraction of *all* U.S. professors were trained in Europe?" it would be appropriate for that person to be included only once; interdisciplinary faculty may be included in multiple calculations, but should never be double-counted in any single calculation.

As an aside: in our preliminary analyses for this study, we considered the alternative: restricting departments (and thus, each individual with them) to one and only one field. However, this required us to make uncomfortable, hard-to-justify, and ultimately counterintuitive choices for departments such as Physics & Astronomy, Computer Science & Engineering, and Biochemistry.

At present, our reasoning and procedure are explained in the Supplement under "Taxonomization of Employment Records." However, we now bring some of this reasoning forward in our rewrite of the Data section of the main text.

The text and figures on up/down mobility could use greater clarification. For example, the text states that, for the typical professor who moves down in the prestige hierarchy, they fall 46.3% downward in Classics & Classical Languages. However, Table S2 suggests that professors in Classics & Classical Languages have downward movement at 41%, on average. Is 46% the high end of that range rather than the average? This should be clarified in the text and Table S2. Furthermore, the statement "while the typical professor who moves upward, of whom there are very few, ascends between 6.5% (agronomy) and 26.1% (Classics and Classical Languages) of the hierarchy. This is also confusing. It is unclear whether this number should represent the proportion of those who move upwards or the rank difference traversed. Neither, however, match Table S2, which suggests that only 4% of Classics & Classical Literature professors have upward mobility and, when they do, they tend to only have a 10% difference in rank. Clarification on these differences would be helpful.

We thank the reviewer for this careful check and comparison (text vs Table S2) as it surfaced the fact that Table S2 had not been updated. This has been fixed, and so the numbers in the main text and the supplement will now correctly agree.

Our intention in the text was to describe the rank differences being traversed among those who move up/down, not percentages of faculty who move in a particular direction. That is, the numbers reported are conditioned on the direction of the move. This has been clarified in the text.

It would be useful to bring in more contextual information to understand the results. For example, might some of the explanation in Medicine & Health be due to the context of university hospitals and practices of soft-money hiring for assistant professors? Could the dominance of Nursing in the Medicine & Health category be an explanation for the gender differences observed in this field?

While we have attempted to bring additional context into the Discussion section of the paper, e.g. gender differences and the *expectations of brilliance* work (Leslie & Cimpian et al, *Science*, 2015), our work lays the necessary foundation for future studies to more deeply analyze the mechanisms that explain why we see the stark patterns and differences we report. To that end, we have:

- Made as much data available as possible, up to the limits provided by our data provider, which will facilitate future work on these and many other questions.
- Amended Supplementary tables with information on the relative sizes of the fields in each domain, and the sizes of each domain.

This allows us to explain, for instance, that while Nursing is the largest among fields in Medicine & Health, constituting 14.5% of Medicine & Health faculty, it is far from dominant. Furthermore, many fields beyond Nursing within Medicine & Health show clear gender differences — women are self-hired significantly more than men in 13 of 14 fields, and in 13 of 14, women also have significantly less mobility than men. Unfortunately, we do not have individual-level information about soft- vs hard-money employment, so we are unable to test the explanation directly.

Prestige appears to be the central concept in this text. However, I cannot find a single definition of prestige or a description in how it was measured in this analysis. I would encourage the authors to include this explicitly in the Data section.

We now directly address this definition in the Data section. We also believe that the paper's scope is somewhat wider now, placing less relative emphasis on prestige.

Academic rank is introduced in the last paragraph of Results. However, there is an important difference noted for Assistant Professors that bears greater examination and discussion. As the authors note, faculty production inequality is universally lower for junior faculty. That is, there are a wider array of institutions producing junior faculty. This is essential for the narrative, which is about hiring networks. In effect, what we are seeing here is that the most pronounced inequity is not in hiring, but in promotion and tenure. **The higher concentration of affiliations at the senior level suggests that attrition diminishes institutional diversity [boldface added by authors]**. One might argue that this entire analysis should only be at the Assistant level, if focused on direct hiring links. At present, the obscuring of the hiring pathway is incredibly problematic—as the authors acknowledge, we do not know whether self-hires are returnees or whether those with upward mobility traversed across several institutions first. The arguments around “academic inbreeding” do not hold if there was high mobility before returning to the home unit.

We thank the reviewer for these insights, as they led directly to our request for additional data, longitudinal analysis, analyses of attrition, etc. We have **boldfaced** what we believe to be an important insight, which is now reflected throughout the paper, in the following concrete ways:

- Fig. 7 (Assistant Professors vs Senior Faculty) is now deleted. Its insights are clarified and replaced by more appropriately stratified analyses.
- New Fig. 3. Our analyses of the Gini coefficients of faculty production are now stratified by *new hires*, defined as those we observe entering the dataset at the Assistant Professor level (meaning, they were hired 2011-2020), and *existing faculty*, defined as the complement of new hires.
- New Fig. 4. Our analyses of gender are now stratified by *new hires* (as above), and *attritions*, defined as those we observe exiting the dataset at any level—thus including all-cause attrition including retirement. This reveals that the composition of the professoriate is moving toward gender parity over time, primarily driven by the retirement of progressively more male-heavy senior cohorts, but *not* driven by dramatic changes in hiring over the past 10y.

These observations, in turn, led us to ask what else may be driven by differential attrition, leading to numerous other new analyses, including:

- New Figs. 1C and 1D. We show that faculty with non-U.S. doctoral training leave U.S. academia at higher rates than U.S.-trained faculty, across all institutions, but that this effect is isolated to those trained abroad *outside* the U.K. and Canada.
- New Fig. 3C. We show that annual risk of attrition is higher for faculty with doctoral degrees from institutions that produce relatively few faculty. This process *exacerbates* observed production inequalities over time, because faculty from institutions with large faculty alumni groups tend to persist longer in academia.
- New Fig. 5B. We show that self-hires leave U.S. academia at higher rates than non-self-hires, across all institutions.

For completeness, we also examined whether mid-career moves, which partially “re-wire” the faculty hiring network, might explain some of the observed time-longitudinal or career-longitudinal effects; they did not.

At present, most of the analyses on gender and rank demonstrate that women make up a larger proportion of the Assistant rank. This is well observed. What is not evident, is how much hiring and institutional prestige factor in the attrition of women scholars. The authors should remove the focus on composition and interrogate the institutional factors as relevant to gender at the Assistant level as compared to more senior levels.

As noted above, our new analyses treat these questions much more directly. See Fig. 5.

The paragraph that suggests a zero-sum game around diversity policies lacks clarity. First, diversity should be clearly defined—is this institutional diversity, field permeability and interdisciplinarity, or socio-demographic diversity. Furthermore, the example is unclear, how is increasing representation of women in Engineering not an individual field-level effort? The argument here is important and unclear.

We have chosen to remove this point from the revised Discussion for two reasons. First, we have chosen to remove the corresponding cross-field and cross-domain hiring analysis from the paper. Second, we agree that the argument itself was rather unclear and arguably too speculative.

One reason that we chose to remove the cross-field and cross-domain hiring analysis was that the estimates of those rates were all lower bounds. It was impossible for our current data to support exact estimates, or even quantify how far the bound might be from the exact value.

There is relatively little analysis and discussion of non-US PhDs. What is presented is entirely at the level of field, and not institution. Once one controls by field, how does institutional prestige interact with the hiring of out-network PhDs? Furthermore, although the data may not be present to interrogate this, the authors should at least acknowledge the large influx of foreign-born scholars who are educated at the baccalaureate levels and receive PhD training here. This is relevant for the hiring network, given research that suggests foreign-born scholars may exhibit different characteristics in salary and selection in the academic job market.

This suggestion has resulted in multiple concrete changes to the paper’s analyses:

- New Fig. 1. We now more prominently analyze non-U.S. PhDs, both in their prevalence across fields (Fig. 1A), and the geographic regions in which they were trained (Fig. 1B). Combining this with analyses over time, we also see that rates of attrition are higher for these faculty, an effect driven mainly by faculty trained outside the UK and Canada (Figs. 1C and 1D).
- Our analyses throughout the paper now use *all* faculty as the denominator, not just U.S. trained faculty.

The second paragraph is unnecessarily technical. These comments can and should be expressed as conceptualizations in the introduction, to be operationalized in the methods section.

This has been fixed.

How were the 391 “degree-producing U.S. institutions” identified—do the authors mean all doctoral-degree granting institutions? If so, this should be specified.

Yes. We have clarified this point.

The authors define gender as “male/female”, though these terms are usually used for sex; furthermore, they should acknowledge the expansive gender identities beyond these dominant two.

We thank the reviewer for this comment, and have audited the paper to avoid males/females when we mean men/women, and we have acknowledged that this is a restrictive binarization of a complex identity. Further discussion of this point is also included in the Supplement.

How do the authors determine fields that were “too small for analysis”? 4595 faculty seems substantial and exceeds the 1k faculty limits mentioned in the supplemental documents. To avoid exclusion, why not aggregate at the higher domain levels that are presented in the text and figures? Why exclude at the specialty level? This makes fields like social sciences highly underrepresented.

This comment stems from a lack of clarity in our presentation. Specifically, when we wrote that some fields were “too small for analysis”, we meant that the fields were too small to be considered as stand-alone fields, and points in our plots. Importantly, however, the individual faculty in those fields were always included in the higher-level (domain, academia) analyses. Thus, it may be more appropriate to think of those individuals as represented in domain-level and academia-level analyses, but hidden from view due to size when we plot individual fields.

We include a new Supplementary Fig. S2 which shows this explicitly, we quantify these exclusions in Supplementary Table S1, and we attempt to be more clear in the main text as well, to avoid confusion.

We also note that, due to our negotiation of additional data from the AARC, we have also been able to include 5 previously excluded fields.

Several of the figures could be improved for readability. Figure 1.C should be reoriented to avoid having to read sideways. “Academia” should be removed from Figure 2—if the total should be represented, perhaps with a dashed line? Figure 3 should be reoriented to avoid sideways reading. Be explicit about what is meant by “top five” in Figure 6. Consider making Figure 7 vertical to improve readability.

We have taken these suggestions into consideration as we changed (and created anew) the figures of the paper. We are open to any additional suggestions here, and thank the reviewer for these ideas.

I would recommend a few small editorial changes. For example:
“academic and research ecosystem” → “academic and research ecosystems”;
“2013and 2017” → “2013 and 2017”;
avoid double “most” in “most fields tend...most fields employ”;
include the term or symbol for percentages on the last sentence of page 13.

Fixed.

The first line could be strengthened. I would argue that faculty hiring is not indirectly related to teaching, training, and mentoring—these are very directly related. Furthermore, there is a noticeable omission of access and networks here; there is a strong relationship between hiring and access to certain networks and opportunities. I would recommend a stronger and more precise opening statement.

Agreed, and fixed.

The authors have greatly expanded their data and analysis beyond their 2015 Science Advances paper. However, they may want to provide some specific language detailing the difference in both approach and findings between these two papers. Given the similarity in title and abstract, it may not be clear what the fundamental contribution of the second paper is, besides increased sample size.

This is a great point, and we hope that the broad scope of this manuscript’s analysis now further differentiates it from previous work.

Referee 2

Referee #2 (Remarks to the Author):

Dear authors,

This is an interesting and impressive paper, which uses a large data set to analyse faculty hiring in the U.S. Moreover, the approach used appears as well chosen and, to the extent that I am able to judge, the analysis has been done in a methodologically sound manner. Moreover, the presentation is clear, the figures are adequate, and the paper is well-structured. The main finding is that U.S academia – across all fields – is organised in a core-periphery structure where a minority of elite institutions produces a majority of all faculty. The study also finds that one in nine professors are hired by their own doctoral university, which is seen as an indication of “self-hiring” being common. Self-hires are also more prevalent when women are appointed, and among higher-ranked institutions. In my view these findings are important, yet at the same time, they are not really new. The hierarchy of U.S academia, both generally and in specific disciplines, has been studied before, as documented in the reference list. Especially, two studies involving authors of the current manuscript (ref 22 and ref 28), reach similar conclusions. While the data set used 191,000 is considerably higher, the main findings echo the Science advances paper published in 2015. Hence, the study under review here confirms previous research rather than revealing new unexpected patterns.

We thank the referee for these encouraging comments. Still, the observation that this work shares much in common with past work, combined with other suggestions from both reviewers and editor, have led us to make concrete changes to differentiate the current manuscript, and to reveal new (and, we believe, unexpected) patterns that both highlight and clarify a number of underlying causal mechanisms that shape the U.S. professoriate.

Most notably, our expanded data set, which now covers 10y, enables us to expand beyond cross-sectional analyses. Specifically, our revised (and mostly rewritten and reframed) manuscript shows numerous puzzling phenomena, of which we highlight just three:

- Fig. 3. Newly hired faculty come from a more diverse set of institutions than existing faculty, yet this is not due to changes in *hiring*. Instead, we find that the risk of *attrition* is nearly twice as high for faculty trained at institutions that typically produce few faculty, indicating that an asymmetric loss of talent plays an important role in maintaining institutional inequalities.
- Fig. 4. The demographics of the professoriate have changed consistently over the past decade, with women’s representation increasing across all domains at remarkably consistent rates. We now identify two key drivers of this observation: (a) new hires are more likely to be women, and (b) retirements are far more likely to be men. Interestingly, we see no evidence that women’s representation among new hires changed between 2011 and 2020: focused only on the new hires, women’s representation has remained

flat. Thus, longitudinal changes in cross-sectional diversity appear to be working their way through, as the next generation replaces the previous.

- Figs. 1C, 1D, 3C, and 5B. The U.S. professoriate differentially loses (i) faculty trained outside the U.S., Canada, and the U.K.; (ii) faculty who come from institutions that produce relatively fewer faculty; and (iii) faculty who are hired at their doctoral alma mater. These differential rates of attrition thus shape the composition of the U.S. professoriate in meaningful ways in the years after hiring.

Moreover, I find that the paper would make an even stronger impression if the empirical findings could be contextualized and compared more systematically. For example, is it possible to find changes over time? Is inequality rising or diminishing in science? A recent paper (<https://www.pnas.org/content/118/7/e2012208118>) found that inequality in terms of citations is increasing, could the same observation apply to academic recruitment? Furthermore, the unequal distribution of resources and rewards in science has been studied for a long time (e.g. Merton) and the distribution found here (that 20 % of the institutions contribute to 80% of the faculty produced) has been seen found also in terms of the production of papers (20% of the authors produce 80% of all scientific papers). Hence, it might be useful to discuss your very interesting findings in relation to how the Pareto principle can be applied to science more generally.

We thank the reviewer for these comments, as they led directly to our request for additional data, longitudinal analysis, analyses of attrition, etc. Essentially all of the analyses from our first submission have now been put into the context of time.

One key example is that shown in the new Fig. 3A, as described above. This shows that 80/20-type inequalities are weaker at initial hiring, but due to differential attrition (more attrition among faculty from institutions that produce fewer faculty), they tend to strengthen over time. Candidly, we believe (though cannot directly test) that this is likely due to culture, belonging, social capital, and tacit knowledge effects. Specifically, this line of argument would suggest that those who are trained at institutions that train more future faculty also lead to those future faculty feeling more “at home” and having more social capital in their new jobs. Other work by our research team, which we disclose has been accepted for publication at *Nature Human Behaviour*, supports this argument as well, showing that more elite U.S. faculty are more likely to have parents who, themselves, are faculty or have a doctoral degree.

In addition to the changes in our analysis, we have revised our Discussion to better include these potential explanations.

Another way of discussing these results would be to reflect on what they say about the structure of U.S Science. Or more concretely, do these findings reflect a more general structure in science or are it due to specific patterns in an American context. In my experience would, for example, the levels of self-hiring found here to be seen as very low in a European context.

We have expanded our discussion of this topic, particularly in light of Fig 1B, which shows that even among U.S. faculty trained outside the U.S., the majority of North American- and European-trained faculty come from Canada and the U.K. Consequently—and in line with the reviewer's hypothesis—we believe that language may play a key role in governing access to the U.S. faculty market, although our data do not directly support a test of such a hypothesis (vs alternative competing hypotheses correlated with language). As a consequence of the size of the U.S. faculty market, there are relatively more opportunities for *non*-self hiring; this may not be the case for smaller language-defined markets, e.g. the Italian or Korean faculty hiring markets.

Overall, I find that contextual comparisons would open for conclusions that more directly could engage, and contribute to, current debates about inequality, discrimination of certain groups, etc. in academia. This would make the paper even more relevant to a large group of researchers and policymakers interested in how science is structured and organised.

These comparisons are generally hard, precisely because there exist no past studies like this one, nor are there equivalent studies in Europe, Africa, Asia, or the Americas.

Nevertheless, we believe that our revised study's analysis of both hiring *and* attrition effects, may directly inform efforts to diversify the U.S. professoriate along axes of prestige, country of origin, and gender. The fact that we observe markedly different rates of attrition by these key variables suggests that there are many points of contact during a career in which one might hope to effectively intervene. We now include these points explicitly in our revised Discussion.

Reviewer Reports on the First Revision:

Referees' comments:

Referee #1 (Remarks to the Author):

I appreciated the opportunity to re-review the manuscript, "Quantifying hierarchy and dynamics in US faculty hiring and attrition". The new additions greatly enhanced the manuscript. I only have a few minor comments that might be useful to address prior to publication.

I would recommend that the authors undergo one more round of editing to improve the precision of the language and remove errors (e.g., not \diamond note). The writing is could be tightened to appeal to a multidisciplinary audience.

The authors report that self-hires are more likely to be women and that self-hires are more likely to leave at higher rates. This might point to temporary, rather than permanent employment. Does Academic Analytics properly distinguish between Visiting Assistant Professors and Assistant Professors?

What is the explanation for some of the unexpected among your prestige ranks? For example, some might find it surprising that Jewish Theological Seminary ranks above UC Davis.

Furthermore, what is # 27 Alfred? Should some sort of threshold be put in place for the prestige ranking? Might these be artifacts of an extremely low relative n ?

In Figure S5: why is the general category (Academia) lower than all the more specific categories? Also, the colors will be indistinguishable for those who are color-blind (here and in other figures where the greens have very similar shading).

Referee #2 (Remarks to the Author):

Dear authors,

Thank you for the exemplary and comprehensive documentation on the changes made. I find that you have addressed the comments by both reviewers in a very constructive manner. Especially important in my view is that additional data has been added, and making this open for others to search is an excellent idea. I have no further comments at this point.

Congratulations on an impressive paper.

Author Rebuttals to First Revision:

Referees' comments:

Referee #1 (Remarks to the Author):

I appreciated the opportunity to re-review the manuscript, "Quantifying hierarchy and dynamics in US faculty hiring and attrition". The new additions greatly enhanced the manuscript. I only have a few minor comments that might be useful to address prior to publication.

I would recommend that the authors undergo one more round of editing to improve the precision of the language and remove errors (e.g., not → note). The writing could be tightened to appeal to a multidisciplinary audience.

We have attempted to tighten the writing without loss of precision throughout the manuscript. We have also edited the not/note typo directly.

The authors report that self-hires are more likely to be women and that self-hires are more likely to leave at higher rates. This might point to temporary, rather than permanent employment. Does Academic Analytics properly distinguish between Visiting Assistant Professors and Assistant Professors?

Academic Analytics differentiates between tenure-track faculty (Assistant, Associate, Full) and non-tenure-track faculty: Lecturers, Instructors, and "Other" faculty. We did not include Lecturers, Instructors, or Other faculty in our analysis. Nevertheless, if some institutions record their Visiting Assistant Professors as simply Assistant Professors when providing data to Academic Analytics, such temporary faculty could, in principle, be included in our sample frame. However, during hand-checking of a subset of AARC data (a few hundred faculty), we observed no Visiting Assistant Professors, and all recorded data were correct.

What is the explanation for some of the unexpected among your prestige ranks? For example, some might find it surprising that Jewish Theological Seminary ranks above UC Davis. Furthermore, what is # 27 Alfred? Should some sort of threshold be put in place for the prestige ranking? Might these be artifacts of an extremely low relative n?

The reviewer's hypothesis is correct — the prestige rankings do not require a particular faculty size to be included. Thus, institutions like the Jewish Theological Seminary and Caltech are small, but elite. However, to avoid confusion, we now include columns in our Extended Data Table that show the faculty size (n faculty) and the number of fields in which that institution has a unit (n departments).

In Figure S5: why is the general category (Academia) lower than all the more specific categories?

The reason for this is that the network itself is more dense with connections, meaning that on average, nodes tend to be closer to each other in geodesic distances. In other words, this phenomenon is simply a consequence of the network density.

Also, the colors will be indistinguishable for those who are color-blind (here and in other figures where the greens have very similar shading).

We have added direct labels on graphs where possible to avoid confusion. We are open to suggestions from the Nature editorial/graphics team.

Referee #2 (Remarks to the Author):

Dear authors,

Thank you for the exemplary and comprehensive documentation on the changes made. I find that you have addressed the comments by both reviewers in a very constructive manner. Especially important in my view is that additional data has been added, and making this open for others to search is an excellent idea. I have no further comments at this point.

Congratulations on an impressive paper.

We thank the reviewer for constructive and encouraging comments and suggestions!